# Developing a multimodal therapy for glioblastoma using oncolytic virus delivering CD19 and EGFRvIII antigens and bi-specific CARs

Jia Li[1], Shyambabu Chaurasiya[2], Guihua Sun [1], Qi Cui[1], Peng Ye[1], Yue Qin [1], Tao Zhou[1], Xiuli Wang [3], Yuman Fong [2], Marcela V. Maus [4,5] & Yanhong Shi [1] ✉

Glioblastoma is the most aggressive primary brain tumor with no cure, largely because of tumor heterogeneity and immunosuppressive tumor microenvironment. Chimeric antigen receptor (CAR)-T cell therapy is highly effective in blood cancers but exhibits limited efficacy in glioblastoma due to heterogeneous tumor antigen expression, antigen loss and poor persistence of tumor-targeting immune cells in glioblastoma. Here we show a multimodal immunotherapy strategy that integrates engineered immune cells with oncolytic viruses to overcome these barriers. We have developed bispecific CAR-T and CAR-NK cells in combination with oncolytic virus that delivers two tumor antigens to glioblastoma cells for effective CAR targeting. Moreover, oncolytic virus armed with membrane-bound interleukin-15 and interleukin-21 enhances immune cell expansion/persistence and cytotoxic activity. This combined approach improves anti-tumor efficacy in vitro and in vivo by limiting immune escape and enhancing anti-tumor immunity. Together, these findings establish a promising platform for multimodal immunotherapy targeting glioblastoma and other solid tumors.

Glioblastoma (GBM) is the most aggressive and refractory primary brain tumor with no cure. Current standard of care treatment for GBM includes surgical resection, radiation and chemotherapy, which only provide limited benefit[1]. Despite advancements in surgical resection, radiotherapy, and chemotherapy, the prognosis for patients with GBM is dismal. Median survival for newly diagnosed GBM patients is still less than 2 years[2] and for recurrent GBM patients is less than 1 year, with even poorer survival when the disease recurs multifocally[3,4], underscoring the urgent need for developing novel therapeutic strategies for GBM treatment[5,6].

Chimeric antigen receptor (CAR) T cell therapy has revolutionized the paradigm of cancer treatment, providing a promising approach to treat hematopoietic malignancies, for which they have become the standard of care. While CAR-T treatment of hematologic malignances has achieved significant success in patients with refractory B cell malignancies, the outcome of this strategy in patients with solid tumors, especially GBM, has not always met with success. CAR-T cells targeting the GBM-specific and associated antigens, including epidermal growth factor receptor deletion mutant variant III (EGFRvIII)[7,8], human epidermal growth factor receptor 2 (Her2)[9], iterleukin-13 receptor alpha 2 (IL13Ra2)[10,11], erythropoietic-producing human hepatocellular carcinoma A2 (Apha2)[12], and GD2 disialoganglioside[13], have been evaluated in phase 1 studies. These CAR-T cells have

[1]Department of Neurodegenerative Diseases, Beckman Research Institute of City of Hope, 1500 E. Duarte Rd., Duarte, CA, USA. [2]Department of Surgery, City of Hope, 1500 E. Duarte Rd., Duarte, CA, USA. [3]Department of Hematology & Hematopoietic Cell Transplantation, City of Hope, 1500 E. Duarte Rd., Duarte, CA, USA. [4]Cellular Immunotherapy Program Cancer Center, Massachusetts General Hospital, Boston, MA, USA. [5]Harvard Medical School, Boston, MA, USA. ✉e-mail: yshi@coh.org

demonstrated acceptable toxicity profiles. However, the outcome of CAR-T treatment of GBM has not been consistent, presumably due to heterogeneous tumor antigen expression, antigen loss or escape following treatment with CAR-T cells targeting a single antigen, and the immunosuppressive tumor microenvironment[7,11,14–17]. Developing strategies to overcome these challenges is needed to achieve robust and consistent effects of CAR-T treatment of GBM.

An important element in CAR-T cell therapy development is the identification of antigens that are confined to tumors. EGFR-vIII is the most common variant of EGFR observed in human tumors[18]. It is expressed in about 30% to 50% of newly diagnosed GBM cases. EGFRvIII results from the in-frame deletion of EGFR exons 2 to7 and the consequent generation of a new amino acid residue glycine when exon 1 links to exon 8. This arrangement creates within the extracellular domain of the EGFR a tumor-specific, immunogenic neoepitope, that is specifically expressed in tumor cells but not in normal cells. Previous studies have demonstrated the target specificity of EGFRvIII CAR and its efficacy against EGFRvIII-expressing tumor cells in vitro and in xenograft mouse models[19,20]. In the first-in-human clinical trial, EGFRvIII CAR-T cells reduced EGFRvIII expression on tumor cells in GBM patients, but there is no clear tumor regression as revealed by MRI imaging due to the emergence of EGFRvIII-negative tumor cells[7]. Designing bi-specific CAR-T cells that target EGFRvIII and other antigens expressed on GBM cells is needed to develop effective EGFRvIII-based CAR-T therapies for GBM.

CD19 has been an excellent target for CAR-T cells against hematological malignancies because of its highly restricted expression on B cells[21]. Studies showing favorable outcome and safety profile using CD19 CAR-T cells have led to the FDA approval of CAR-T therapies for B cell malignancies[22,23]. We hypothesize that the ectopic expression of CD19 on GBM tissues could allow bi-specific CAR-T cells that target both the GBM-specific EGFRvIII and the well-characterized CD19 to target GBM effectively.

OV represents a versatile platform for targeting tumors via selective replication and lysis within malignant cells[24–26]. In addition to tumor cell lysis, OV also activates the host immune response, which can stimulate antigen outspread and cytokine release[27]. OV exhibits immune-stimulating effects and can turn the immunologically 'cold' tumor microenvironment to the 'hot'. Therefore, OV has been included in combinatorial therapies to boost the efficacy of cancer immunotherapy[28]. Furthermore, because these viruses target tumor cells selectively, they can be used as vectors to deliver transgenes to tumor cells specifically[27]. Indeed, OVs can be engineered to express tumor-associated antigens, transforming infected tumor cells into antigen-presenting cells[29]. Moreover, OV has been genetically engineered to express cytokines/chemokines to enhance the recruitment of tumor-targeting immune cells or promote the effect of immune checkpoint inhibitors, thus strengthening antitumor efficacy[30]. The first clinically approved OV, talomogene laherperepve (TVEC), is a genetically modified herpes simplex virus (HSV) that expresses the granulocyte-macrophage colony-stimulating factor (GM-CSF) cytokine[31].

Because of its ability to promote the proliferation, persistence, and cytotoxicity of natural killer (NK) cells[32,33], interleukin (IL)−15 has been shown to enhance the activity of CAR-NK cells in both preclinical and early clinical studies[34–38]. IL-21 is another cytokine that is appealing to cancer immunotherapy, because it can not only promote mitochondrial biogenesis and metabolic fitness in T cells[39] but also support the expansion and functionality of NK cells[40–43]. Indeed, a recent study showed that NK cells armed with IL-21 exhibit favorable safety and antitumor efficacy against GBM[44]. However, whether one can equip OVs with both IL-15 and IL-21 to enhance cancer immunotherapy remains tantalizing to be tested.

In addition to CAR-T cells, CAR-NK cells have emerged as a promising approach of adoptive cancer immunotherapy to complement CAR-T approach[45], especially for solid tumors, for which CAR-T has not exhibited consistent success. Because NK cell cytotoxicity is not dependent on human leukocyte antigen (HLA) matching and does not have the risk of graft versus host disease (GvHD), NK cells can be developed into allogeneic, "off-the-shelf" cell therapy[46]. NK cells can be obtained from multiple cell sources. Because human pluripotent stem cells (hPSCs) are self-renewable, they can serve as source cells to produce unlimited number of NK cells for cell therapy[47–50].

To address the major challenges faced in cancer immunotherapy for GBM, including heterogeneous tumor antigen expression, antigen loss or escape, and the immunosuppressive tumor microenvironment, in this study, we developed a multimodal approach to target GBM. We engineered dual-antigen oncolytic vaccinia virus (OVDual) encoding truncated CD19 (CD19t) and EGFRvIII (EGFRvIIIt) to deliver the CD19 and EGFRvIII antigens to GBM tissues. In parallel, we generated bispecific CAR-T (BiCAR-T) cells targeting both CD19 and EGFRvIII antigens. To further improve the therapeutic potential, we engineered oncolytic vaccinia virus encoding both mIL-15 and mIL-21 (OVmIL15/21) and evaluated the effects together with OVDual and BiCAR-T cells. In addition to BiCAR-T cells, we explored the development of BiCAR-engineered NK cells derived from hPSCs as a potential "off-the-shelf" therapeutic strategy for GBM.

## Results

### Engineering OVDual encoding both CD19t and EGFRvIIIt and BiCAR-T cells equipped with both CD19-CAR and EGFRvIII-CAR

Because EGFRvIII is solely expressed on tumor tissues but not normal tissues, it provides a unique tumor-specificity for CAR-T targeting. However, GBM is highly heterogenous. EGFRvIII is only expressed on 30 to 50% GBM tissues. For example, we only detected the expression of EGFRvIII on patient-derived GBM cell line PBT419 but not on patient-derived GBM cell lines PBT726, PBT022, PBT707, PBT017, PBT003 (Figure. S1). To overcome this antigenic heterogeneity, we designed an oncolytic virus encoding a truncated form of EGFRvIII to deliver the EGFRvIII antigen to all infected GBM cells. Because CD19 CAR-T cells have demonstrated a favorable outcome and safety profile and have the FDA approval to serve as a B cell malignancy therapy[22,23], we intended to introduce the human CD19 antigen to GBM tissues using OV so that we can take advantage of the highly effective CD19 CAR-T cells to treat GBM. Accordingly, we generated an oncolytic chimeric orthopoxvirus that encodes both CD19t and EGFRvIIIt driven by the control of the vaccinia synthetic early promoter and modified H5 promoter, respectively. These promoters enable rapid expression of CD19t and EGFRvIIIt prior to oncolytic virus-mediated tumor lysis. The human CD19t (hCD19t) and EGFRvIIIt (hEGFRvIIIt) expression cassette was inserted into the J2R and the F14.5L locus, respectively. This modified OV is called CF17-CD19tEGFRvIIIt, hereinafter referred to as OVDual (Fig. 1A).

To validate the ability of OVDual to infect GBM cells and deliver the CD19t and EGFRvIIIt antigens to the GBM cell surface, patient-derived GBM cells were infected with OVDual at varying multiplicity of infection (MOI) for 24 hours (h), followed by flow cytometry analysis using antibodies specific for the vaccinia viral protein, CD19, and EGFRvIII. We detected the expression of the vaccinia viral protein and the CD19 and EGFRvIII tumor antigens on the surface of GBM cells in a dose dependent manner (Fig. 1B and C). Confocal imaging confirmed the expression of the vaccinia viral protein, CD19, and EGFRvIII on GBM cells at both low and high MOIs (Fig. 1D).

To target both CD19 and EGFRvIII antigens, we designed a lentiviral vector encoding a CD19-EGFRvIII bispecific CAR (BiCAR) for simultaneous targeting of these two antigens. The EGFRvIII CAR was placed upstream of the CD19 CAR and linked by a G4S × 4 linker. Additionally, the vector includes a truncated CD34 (CD34t) as a tag for the BiCAR[51], the CD28 transmembrane and intracellular domains, and the CD3 signaling moiety (Fig. 1E). Human peripheral blood

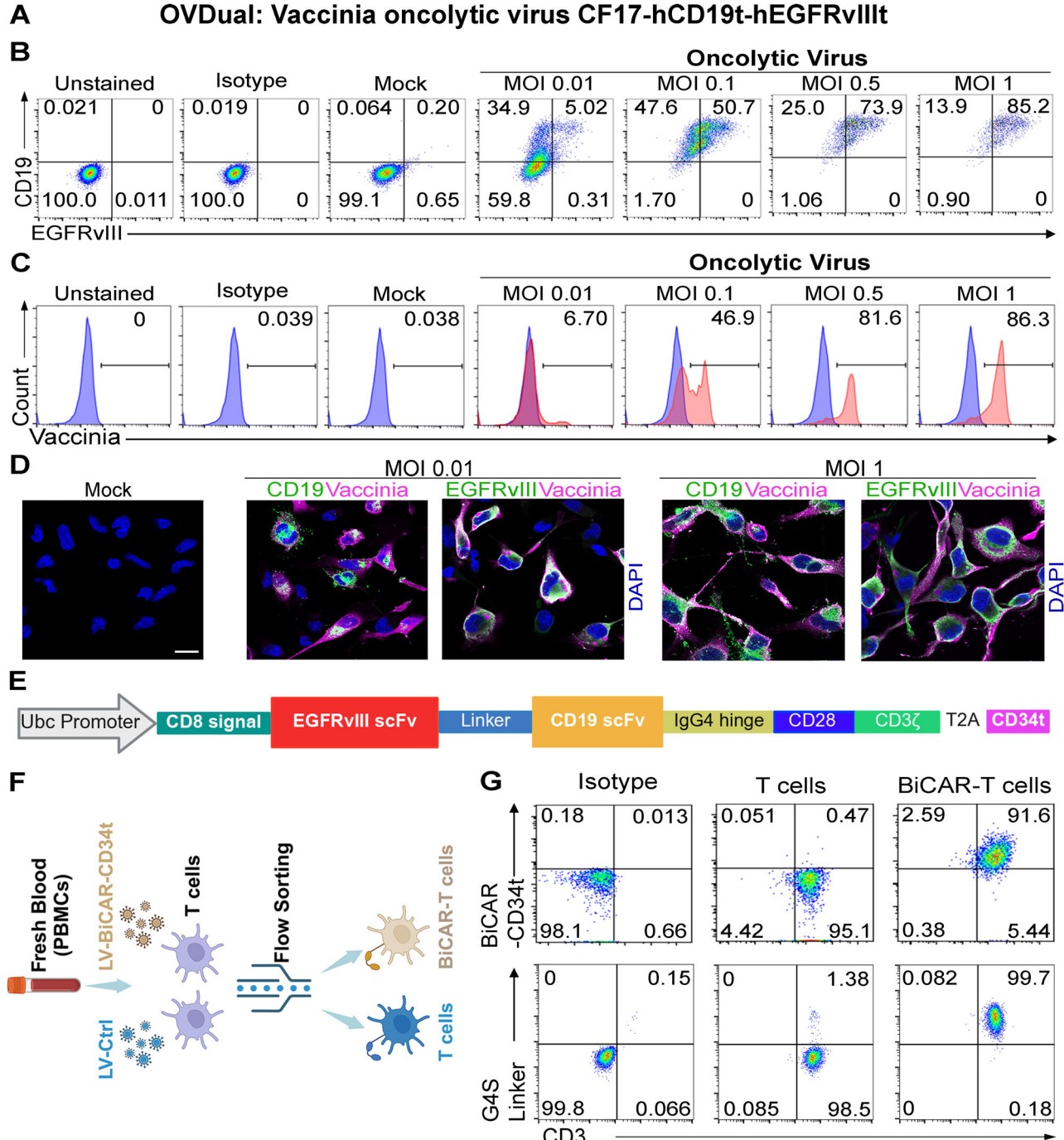

**Fig. 1 | Generation of dual-antigen-encoding OV and bispecific CAR-T cells.**
**A** Defining CF17-hCD19t-hEGFRvIIIt as OVDual. **B** Representative FACS plots of GBM cells positive for both CD19 and EGFRvIII after 24 h of OVDual infection at increasing MOIs. The percentage of CD19-positive and EGFRvIII-positive population is indicated in relevant quadrants. **C** Representative FACS plots of GBM cells positive for vaccinia viral protein after 24 h of OVDual infection at increasing MOIs. The percentage of vaccinia virus-positive population is indicated in the graph.

**D** Representative images of GBM cells infected with OVDual at a MOI of 0 (mock), 0.01 or 1. Scale bars, 20 μm. DAPI (4′6-diamidino-2-phenylindole) counterstaining is shown in blue. Representative image is shown from *n* = 2 independent experiments. **E** A schematic of the BiCAR construct (generated using BioRender). **F** A schematic of BiCAR-T generation. **G** Representative FACS plots of BiCAR-T and control T cells using anti-CD34, anti-G4S linker and anti-CD3 antibodies.

mononuclear cells (PBMC) were transduced with the lentiviral vector of BiCAR (LV-BiCAR-CD34t) to generate BiCAR-T cells. The transduced cells were sorted by flow cytometry using antibodies for CD3 and CD34 and followed by expansion (Fig. 1F). The percentage of the CD3 and G4S linker double-positive cells as well as CD3 and BiCAR double-positive cells in the expanded BiCAR-transduced T cells was evaluated and more than 90% of cells were CD3/G4S linker-double-positive and CD3/BiCAR-double-positive (Fig. 1G).

## BiCAR-T cells target OVDual-infected GBM cells effectively in vitro

To determine whether BiCAR-T cells can specifically target OVDual-infected GBM cells, patient-derived GBM cells (PBT003) were first infected with OVDual for 5 h and subsequently cocultured with BiCAR-T cells. Cell viability was assessed using a standard killing assay. Compared with BiCAR-T cells cocultured with OVGFP-infected GBM cells, OVDual-infected GBM cells exhibited significantly reduced

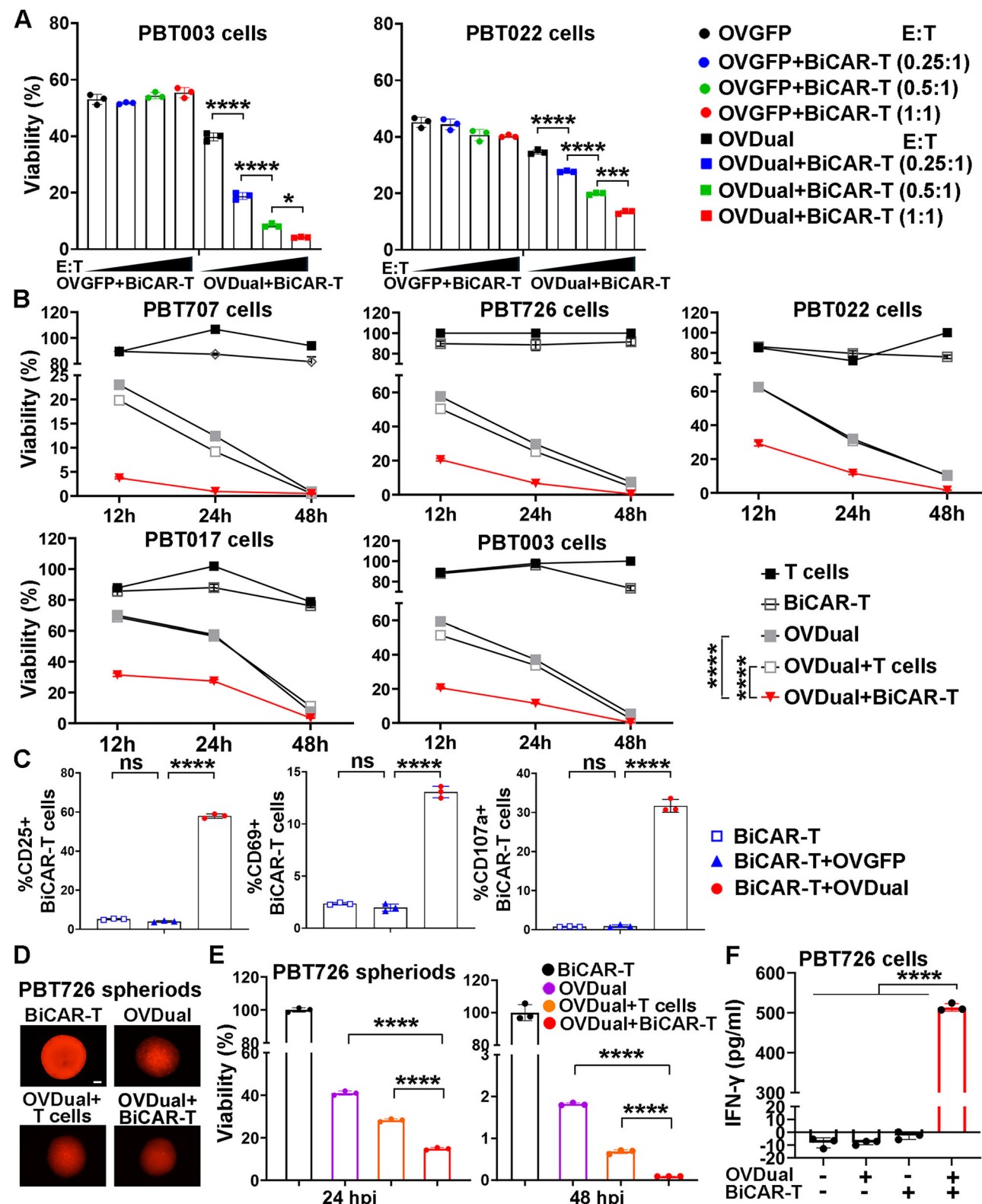

viability in a dose-dependent manner as the effector-to-target (E: T) ratio increased (Fig. 2A). BiCAR-T also triggered effective tumor killing in response to either CD19 or EGFRvIII antigen alone (Figure. S2A and B). These results demonstrate that BiCAR-T cells can recognize both CD19 and EGFRvIII antigens and eliminate GBM cells in an antigen-dependent manner. Moreover, we detected substantially more potent killing of GBM cells infected with OVDual and cocultured with BiCAR-T cells, compared to that of GBM cells infected with OVDual but

cocultured with control T cells not equipped with CAR or not cocultured with any T cells (Fig. 2B and Figure. S3). Accordingly, BiCAR-T cells combined with OVDual demonstrated superior tumor-killing ability compared to control T cells with OVDual. In contrast, control T cells or BiCAR-T cells cocultured with GBM cells that were not infected with OVDual exhibited minimal GBM cell killing (Fig. 2B and Figure. S3). Flow cytometry analysis revealed that BiCAR-T cells cocultured with OVDual-infected GBM cells (BiCAR-T+OVDual)

**Fig. 2 | Anti-tumor efficacy of combination therapy of OVDual and BiCAR-T cells in vitro. A** BiCAR-T–mediated cytotoxicity was measured in PBT003 and PBT022 GBM cells infected with OVDual using a luciferase reporter assay. Infected GBM cells were co-cultured with BiCAR-T cells for 24 h. $n = 3$ cell culture replicates. Data are represented as mean ± SD and were analyzed by one-way ANOVA test with Tukey's multiple comparison test. $*p = 0.0112$, $***p = 0.0002$, and $****p < 0.0001$. **B** GBM cells (PBT707, PBT726, PBT022, PBT017, PBT003) were infected ± OVDual (MOI = 1, 5 h) and then treated ± control T or BiCAR-T cells at an E/T ratio of 1. Cell viability was measured by luciferase assay at 12, 24, and 48 h after control T or BiCAR-T treatment. $n = 3$ cell culture replicates. Data are represented as mean ± SD and were analyzed by two-way ANOVA with Tukey's multiple comparison test. $****p < 0.0001$. **C** Enhanced activation of BiCAR-T cells cocultured with PBT003 cells infected with OVDual (BiCAR-T+OVDual) as revealed by flow cytometry, compared

to BiCAR-T cells cocultured with PBT003 cells infected with OVGFP (BiCAR-T + OVGFP) or uninfected PBT003 cells (BiCAR-T). $n = 3$ cell culture replicates. Data are represented as mean ± SD and were analyzed by one-way ANOVA test with Tukey's multiple comparison tests. $****p < 0.0001$ and ns means no significant difference. (**D** and **E**) 3D tumor spheroids of RFP-labeled PBT726 cells were infected with OVDual and then treated with BiCAR-T cells. The size of tumor spheroids is shown in (**D**) and cell viability in tumor spheroids is shown in (**E**). Scale bar, 100 μm. $n = 3$ cell culture replicates. Data are represented as mean ± SD and were analyzed by two-tailed unpaired t tests. $****p < 0.0001$. (**F**) IFNγ production was assessed in conditioned medium from PBT726 cells infected ± OVDual and treated ± BiCAR-T cells. $n = 3$ cell culture replicates. Data are presented as mean ± SD and were analyzed by one-way ANOVA with Tukey's multiply comparison test. $****p < 0.0001$. Source data are provided in the Source Data file.

exhibited markedly increased expression of activation markers CD25, CD69, and CD107a, compared to BiCAR-T cells cocultured with OVGFP-infected GBM cells (BiCAR-T + OVGFP) or BiCAR-T cells cocultured with uninfected GBM cells (BiCAR-T) (Fig. 2C). BiCAR-T cells cocultured with OVCD19t- or OVEGFRvIIIt-infected GBM cells also expressed increased level of activation markers CD25, CD69, and CD107a, compared to control BiCAR-T co-cultured with OVGFP-infected or uninfected GBM cells (Figure. S2C and D), These results together indicate that BiCAR-T cells can be activated in response to OV-delivered single or dual antigens to eliminate tumor cells in vitro.

To mimic the three-dimensional (3D) cellular context in GBM tumor tissues, we generated GBM spheroids. Enhanced tumor killing by combining OVDual and BiCAR-T cells was also detected in 3D GBM spheroids, similar to the effect on 2D GBM cells. We detected much more potent killing of OVDual-infected 3D GBM spheroids treated with BiCAR-T cells, compared to control T cells without BiCAR or no T cells (Fig. 2D and E).

Next, we evaluated the capacity of BiCAR-T cells to release cytokines when encountered with GBM cells infected with or without OVDual. The OVDual and BiCAR-T combination treatment group exhibited a significantly higher concentration of IFNγ compared to either the OVDual alone or BiCAR-T alone treatment group (Fig. 2F). Taken together, these data indicate that OVDual can deliver antigens that are both GBM-specific and not normally GBM-expressed, in this case EGFRvIIIt and CD19t, to GBM cells, and induce antigen-specific CAR-T cell-mediated anti-tumor activity. Moreover, we detected enhanced migration of BiCAR-T cells toward OVDual-infected GBM cells, compared to that of control T cells, in a transwell system (Figure. S4). Taken together, these results indicate that BiCAR-T cells exhibit potent efficacy again GBM cells when combined with OVDual.

### Anti-tumor efficacy of the combination therapy of OVDual and BiCAR-T cells in a xenograft GBM mouse model

To evaluate the anti-tumor activity of OVDual and BiCAR-T combination therapy in vivo, we established an orthotopic xenograft GBM mouse model by transplanting a luciferase reporter-expressing GBM cells (PBT003-luc) intracranially into the frontal lobe of NOD/SCID/IL-2rg (NSG) mice. One week after GBM cell transplantation, mice were treated with OVDual intratumorally, followed by an intratumoral injection of BiCAR-T cells or vehicle control two days after. A second dose of OV was injected into the OVDual alone or OVDual+BiCAR-T treatment group to boost the anti-tumor effect (Fig. 3A). Tumor progression was monitored by weekly bioluminescence imaging. Consistent with our in vitro observation, tumor-bearing mice treated with the combination of OVDual and BiCAR-T cells showed much more reduced tumor progression compared to no-treatment, BiCAR-T alone treatment, or OVDual treatment alone control mice (Fig. 3B, C). For example, at week 1, the percent reduction of tumor luciferase activity relative to no-treatment control was -0.58% for BiCAR-T, 78.47% for OVDual alone, and 94.5% for OVDual+BiCAR-T group; and at week 2, the percent reduction of tumor luciferase activity relative to no-

treatment control was 12.5% for BiCAR-T, 79.64% for OVDual alone, and 95.96% for OVDual+BiCAR-T group. Importantly, tumor-bearing mice treated with the combination of OVDual and BiCAR-T exhibited dramatically prolonged survival than control mice without treatment or mice treated with BiCAR-T or OVDual alone (Fig. 3D). These results together indicate that the combination therapy of OVDual and BiCAR-T cells is more effective in the inhibition of tumor progression compared to BiCAR-T or OVDual alone therapy.

Consistent with in vitro data, treatment with OVCD19t or OVEGFRvIIIt in combination with BiCAR-T cells also led to a marked reduction in tumor volume as measured by bioluminescence imaging, compared to OVGFP combined with BiCAR-T cells (Figure S5A–C). Accordingly, mice received OVCD19t or OVEGFRvIIIt plus BiCAR-T cells survived longer than mice treated with OVGFP and BiCAR-T or received no treatment (Figure S5D). These findings indicate that both the CD19CAR and EGFRvIIICAR components of BiCAR-T cells are functionally active in vivo. Moreover, we compared OVDual plus BiCAR-T cells with OVEGFRvIIIt plus EGFRvIII CAR-T cells (Figure. S6A and B). Mice received BiCAR-T cells together with OVDual exhibited a modest survival advantage ($p = 0.08$) compared to mice treated with OVEGFRvIIIt plus EGFRvIIICAR-T cells (Figure. S6C).

To evaluate the effectiveness of OVDual in delivering CD19 and EGFRvIII antigens to tumor cells, we examined brain tissues from GBM-bearing mice treated with OVDual alone or OVDual in combination with BiCAR-T cells. We transplanted RFP-labeled GBM cells (PBT003-RFP) into the frontal lobe of NSG mice to establish xenografted tumors in the brain, and then evaluated the expression of hCD19 and hEGFR-vIII, or the presence of vaccinia virus and BiCAR-T cells within tumors and surrounding tumor microenvironment following the treatment with OVDual alone, BiCAR-T cells alone, or the combination of OVDual and BiCAR-T cells. After treatment, mouse brains were collected and subjected to cryosection and immunostaining (Figure. S7A). To confirm antigen delivery by OVDual, brain sections from GBM-bearing mice treated with OVDual were immunostained for hCD19 and hEGFRvIII. Strong hCD19 and hEGFRvIII signals were observed within the tumor regions of OVDual-treated mice, indicating efficient antigen delivery by oncolytic virus. In contrast, tumors from mice without OVDual treatment exhibited negligible staining for either antigen (Figure. S7B-D). Notably, in the presence of BiCAR-T cells, an even higher percentage of hCD19+ or hEGFRvIII+ cells was detected in the GBM cell (PB003-RFP) population, suggesting that BiCAR-T cells can facilitate tumor antigen spreading within tumor tissues. Together, these results demonstrate that OVDual can effectively deliver hCD19 and hEGFRvIII antigens to tumor cells in vivo, particularly in the context of BiCAR-T treatment.

Consistent with the detection of OVDual-encoded antigens CD19 and EGFRvIII, we detected vaccinia virus signals in tumor cells from mice treated with OVDual (Figure. S7E and F). Moreover, we detected a stronger vaccinia virus signal in tumor cells from mice treated with the combination of OVDual and BiCAR-T cells, compared to that in mice treated with OVDual alone, suggesting a greater spread of OVDual in

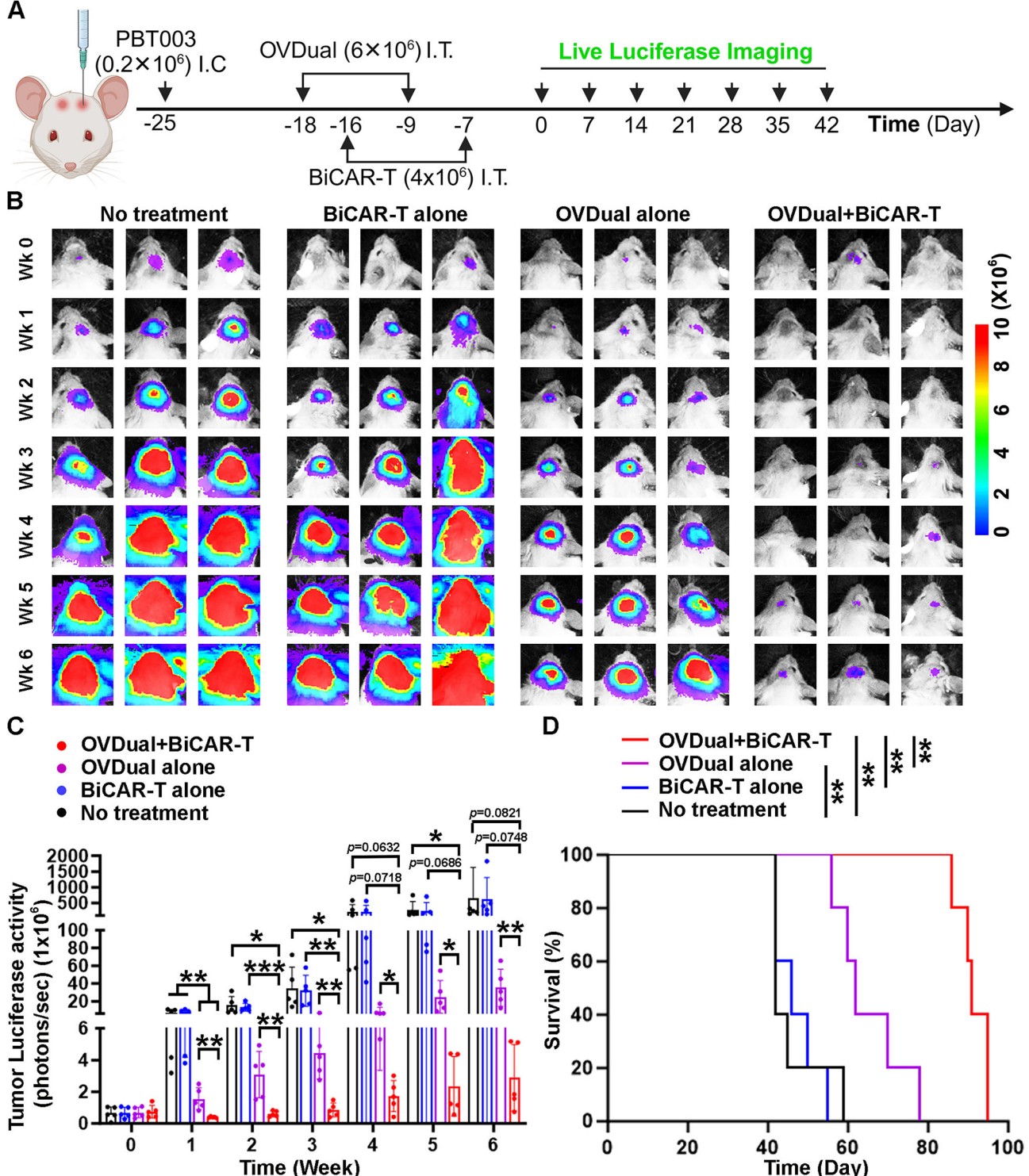

**Fig. 3 | The combination of OVDual and BiCAR-T cells exerts potent anti-tumor efficacy in a xenograft GBM mouse model. A** A schematic of GBM (PBT003-luc) tumor-bearing mice treated with OVDual and BiCAR-T cells. I.C.: intracranial; I.T.: intratumoral (generated using BioRender). **B** Bioluminescence images of brain tumors in NSG mice. Because this experiment was performed concurrently with the experiments for Figure S5 and Figure S6, the no-treatment control group is shared among these figures, and the OVDual+BiCAR-T group in this figure is shared with Figure S6. **C** Quantification of the bioluminescence intensity of tumors after GBM cell transplantation and treatment. $n = 5$ mice per group. Data are shown as mean ± SD and were analyzed by two-tailed unpaired t tests. Week1: **$p$ (OVDual vs no treatment) = 0.0064, **$p$ (OVDual+BiCAR-T vs no treatment) = 0.0020, **$p$ (OVDual vs BiCAR-T) = 0.0039, **$p$ (OVDual+BiCAR-T vs BiCAR-T) = 0.0010, **$p$ (OVDual+BiCAR-T vs

OVDual) = 0.0056. Week2: *$p$ (OVDual+BiCAR-T vs no treatment) = 0.0129, ***$p$ (OVDual+BiCAR-T vs BiCAR-T) = 0.0002, **$p$ (OVDual+BiCAR-T vs OVDual) = 0.0056. Week3: *$p$ (OVDual+BiCAR-T vs no treatment) = 0.0146, **$p$ (OVDual+BiCAR-T vs BiCAR-T) = 0.0040, **$p$ (OVDual+BiCAR-T vs OVDual) = 0.0014. Week4: *$p$ (OVDual +BiCAR-T vs OVDual) = 0.0196. Week5: *$p$ (OVDual+BiCAR-T vs no treatment) = 0.0459, *$p$ (OVDual+BiCAR-T vs OVDual) = 0.0264. Week6: **$p$ (OVDual+BiCAR-T vs OVDual) = 0.0091. **D** The survival of GBM tumor–bearing mice treated with OVDual alone, BiCAR-T alone, OVDual plus BiCAR-T, or no treatment. $n = 5$ mice per group. Data were analyzed by log-rank test. **$p$ (OVDual vs no treatment) = 0.0063, **$p$ (OVDual+BiCAR-T vs no treatment) = 0.0015, **$p$ (OVDual+BiCAR-T vs OVDual) = 0.0018, **$p$ (OVDual+BiCAR-T vs BiCAR-T) = 0.0018. Source data are provided in the Source Data file.

tumor tissues in the presence of BiCAR-T cells, consistent with the observation shown in Figure S7B–D. In contrast, no vaccinia virus signal was detected in brains of mice treated with BiCAR-T alone or received no treatment (Figure. S7E and F). Additionally, increased hCD45-positive cells were detected within tumor tissues in mice treated with OVDual and BiCAR-T cells, compared to that in mice treated with OVDual or BiCAR-T cells alone, indicating enhanced infiltration of BiCAR-T cells in the presence of OVDual (Figure. S7E and G). Notably, the OVDual and BiCAR-T combination treatment resulted in a more dramatic reduction in the tumor size compared to either treatment alone as revealed by the reduced population of RFP-labeled GBM cells (Figure. S7B and E).

### Enhancement of immune cell cytotoxicity and survival by membrane-bound IL15 and IL21 (mIL15/21) in vitro

Cytokine signaling is critical for the anti-tumor activity and survival of cytolytic immune cells including CD8⁺ T cells and NK cells. Because both IL15 and IL21 can support the expansion and function of immune cells individually[32–43], we asked if OV delivering membrane-bound IL15 and IL21 (mIL15/21) together can enhance the anti-tumor activity of immune cells. Prior to answering this question, we first confirmed if combining mIL15/21 can promote immune cell cytotoxicity against GBM cells. To address this, we first generated a lentiviral construct that expresses mIL15/21 on the same vector. The human IL15 sequence is linked to the hinge and transmembrane domains derived from human CD8, and the signaling domain of human CD8, followed by the T2A peptide sequence and then membrane-bound IL21 in the same structure as membrane-bound IL15. GBM cells (PBT003) were transduced with the lentivirus encoding mIL15/21 (LVmIL15/21). Flow cytometry analysis showed that both IL15 and IL21 were well expressed on the surface of GBM cells (Figure. S8A). LVmIL15/21-transduced GBM cells increased the efficacy of OVDual and BiCAR-T cell combination against GBM cells substantially (Figure. S8B). These data together indicate that mIL15/21 is a potent enhancer of BiCAR-T cytolytic effect again OVDual-infected GBM cells.

After confirming the potent effect of LVmIL15/21-transduced GBM cells on enhancing CAR-T cell expansion and cytotoxicity, we next constructed oncolytic viruses encoding mIL15/21. The cDNA fragments encoding mIL15 and mIL21 were inserted into the CF17 poxvirus vector to allow simultaneous expression of mIL15 and mIL21 as we did to generate the OVDual vector, and the resultant OV was termed OVmIL15/21 (Fig. 4A). Subsequently, GBM cells including PBT003 and PBT022 cells were infected with OVmIL15/21 for 24 h, and then the infected cells were collected and subjected to Western blot, immunostaining, and flow cytometry analyses to detect the total protein and membrane protein levels of IL15 and IL21, respectively. Both IL21 and IL15 were detected on OVmIL15/21-infected GBM cells as revealed by Western blot (Fig. 4B). Immunostaining revealed that GBM cells infected with OVmIL15/21 expressed membrane-bound IL15 and IL21 on the cell surface (Fig. 4C). Flow cytometry showed a dose-dependent increase in the percentage of tumor cells positive for mIL15, mIL21 and vaccinia (Fig. 4D), consistent with the observed reduction in tumor viability in response to OVmIL15/21 treatment in a dose-dependent manner (Fig. 4E).

To evaluate the effects of OVmIL15/21 on T cell expansion and cytotoxicity, we added OVmIL15/21 to the combination of OVDual and BiCAR-T treatment and included OVGFP as a control for OVmIL15/21. Infection of GBM cells with OVmIL15/21 together with OVDual promoted the expansion of cocultured BiCAR-T cells significantly, compared to infection of GBM cells with OVGFP and OVDual (Fig. 4F). We further analyzed the levels of secreted IFNγ in conditioned medium under different treatment conditions. We detected a considerably higher IFNγ level in the conditioned medium of BiCAR-T cells cocultured with GBM cells infected with OVmIL15/21 together with OVDual, compared to that when GBM cells were infected with OVDual alone. Moreover, there was an OVmIL15/21-dose-dependent increase of IFNγ secretion in the treatment group of OVmIL15/21 together with OVDual

and BiCAR-T (Fig. 4G). Furthermore, BiCAR-T cells exhibited stronger killing effect against GBM cells infected with OVmIL15/21 together with OVDual, compared to that against GBM cells infected with OVGFP and OVDual (Fig. 4H). For example, % reduction in tumor cell luciferase activity was 97.40% with BiCAR-T+OVDual+OVmIL15/21 treatment vs 89.47% with BiCAR-T+OVDual+OVGFP treatment in PBT022 cells at E: T of 0.5 and 24 h post-treatment ($p < 0.0001$) (Fig. 4H). In contrast, control T cells without CAR showed no significant difference in cytotoxicity against GBM cells infected with OVmIL15/21 or OVGFP together with OVDual (Figure. S9), indicating that the stimulatory effect of OVmIL15/21 requires CAR-T cell and antigen engagement. These results together indicate that OVmIL15/21-infected cells can enhance the expansion and cytotoxicity of BiCAR-T cells in vitro.

To define molecular mechanisms underlying OVmIL15/21-boosted tumor killing effect by BiCAR-T, BiCAR-T cells were cocultured with GBM cells infected with OVDual in combination with OVmIL15/21 or OVGFP. Bulk RNA-seq analysis of BiCAR-T cells collected 48 h after treatment—a time point when tumor cells were eliminated—revealed robust transcriptional change induced by OVmIL15/21. A total of 190 genes were significantly differentially expressed (padj<0.05), with 112 genes upregulated and 78 genes downregulated in BiCAR-T cells treated with OVmIL15/21 compared to BiCAR-T cells treated with OVGFP (Figure. S10A). Pathway enrichment analysis revealed that the upregulated genes were predominantly associated with cell killing, chemotaxis, leukocyte activation, T cell-mediated immunity, immune effector process, chemokine signaling, leukocyte-mediated cytotoxicity, and positive activation of T cell activation (Figure. S10B), all of which could lead to enhanced tumor cytotoxicity by BiCAR-T cells. Further analysis of upregulated genes for reactome pathways revealed a strong enrichment in cytokine signaling pathways, including IL-15, IL-2, IL-3, IL-5, and GM-CSF signaling (Figure. S10C). The activation of IL-15 and IL-21 signaling pathways was further demonstrated by the upregulation of their canonical signaling component JAK3 and the shared receptor subunit IL2RB as shown in the targeted heatmap (Figure. S10D). This cytokine-driven state was further evidenced by the elevated expression of cytotoxic effector molecules PRF1 (perforin 1), GNLY (granulysin), and GZMH (granzyme H)[52–54] in BiCAR-T cells treated with OVmIL15/21 vs OVGFP (Figure. S10D), establishing a functional link between pathway activation and enhanced BiCAR-T cell cytolytic capacity. Together, these results demonstrate that OVmIL15/21 delivers functional membrane-bound cytokines to boost BiCAR-T tumor cytotoxicity by activating pathways related to leukocyte chemotaxis, leukocyte activation, and immune effector process.

### OVmIL15/21 enhances the efficacy of OVDual and BiCAR-T combination therapy in a xenograft GBM mouse model

To determine the effect of OVmIL15/21 in vivo, we transplanted RFP-labeled GBM cells (PBT003-RFP) into the frontal lobe of NSG mice to establish tumor. 7 days after GBM cell transplantation, we administered a fixed dose of OVDual ($3 \times 10^6$ PFU) combined with varying doses of OVmIL15/21 ($3 \times 10^6$ or $6 \times 10^6$ PFU) into transplanted mice, followed by BiCAR-T treatment 2 days after OV administration (Fig. 5A). Mice treated with OVDual plus OVmIL15/21 without BiCAR-T or received no treatment were included as controls. Treatment with OVDual and OVmIL15/21 together with BiCAR-T cells led to a better distribution of OV around the tumor area, compared to the treatment with OVDual and OVmIL15/21 without BiCAR-T cells (Fig. 5B and C), presumably because BiCAR-T-induced tumor cell lysis enhanced the release of OV and infection of surrounding tumor cells by the released OV. Moreover, treatment with OVDual and a higher dose of OVmIL15/21 together with BiCAR-T cells resulted in a broader intratumoral distribution of BiCAR-T cells, as revealed by an increased area of hCD45-positive cells within RFP-positive tumor tissues (Fig. 5D), and accordingly, a smaller tumor size as revealed by the reduced population of RFP-positive GBM cells (Fig. 5B). These results indicate that OVmIL15/

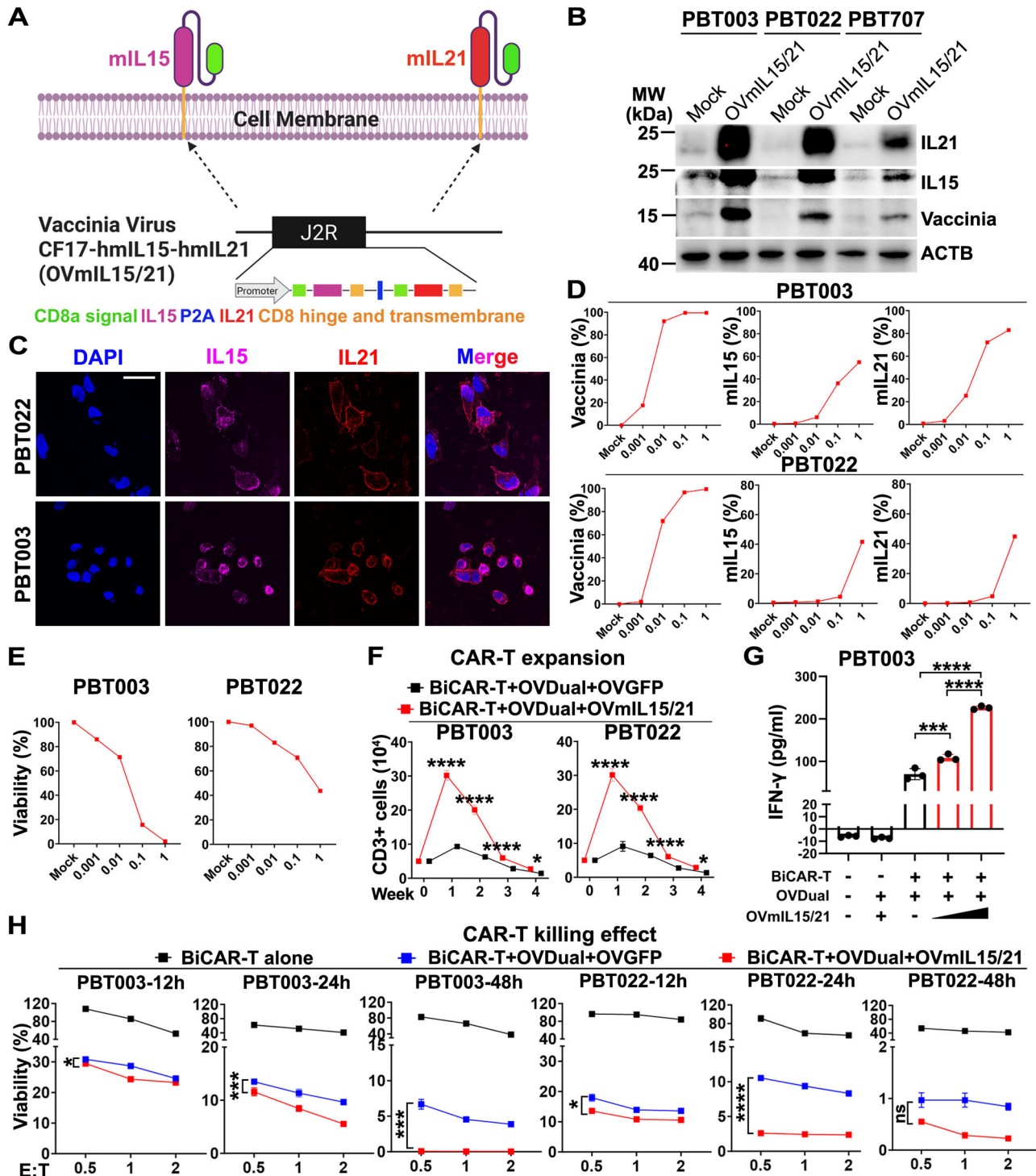

21 can serve as a critical activator of BiCAR-T cell expansion/persistence within the tumor microenvironment, providing a plausible explanation for the enhanced efficacy of OVDual and BiCAR-T combination therapy by OVmIL15/21.

In addition, microglia have been shown to play an important role in modulating the GBM tumor microenvironment[55]. To determine how microglia in the tumor microenvironment were affected by the combination therapy, we analyzed the expression of IBA1, a marker of pan-microglia, and CD68, a marker of reactive microglia, the expression of which reflects immune activation[56], on brain slices from mice subjected to different treatments. Mice treated with the combination of

BiCAR-T cells, OVDual, and OVmIL15/21 exhibited stronger IBA1-positive and CD68-positive signals in the tumor microenvironment compared to control groups (Fig. 5E). Combination treatment including OVmIL15/21 promoted microglial activation and infiltration, as revealed by an increased percentage of IBA1+ cells and CD68 + IBA1+ cells within the RFP-positive tumor tissues (Fig. 5F, G). Together, these results indicate that OVmIL15/21 modulates the tumor microenvironment by enhancing CAR-T cell expansion and persistence while promoting microglial activation and infiltration.

We next tested if OVmIL15/21 can enhance the efficacy of OVDual and BiCAR-T combination therapy in vivo. For this purpose, luciferase

**Fig. 4 | Oncolytic virus encoding membrane-bound (m) IL-15 and IL-21 (OVmIL15/21) enhances BiCAR-T cell expansion and cytotoxicity against GBM cells in vitro. A** A schematic illustration of OVmIL15/21 (generated using BioRender). **B** Protein expression of human mIL15, mIL21, or vaccinia viral protein in OV-infected GBM cells. PBT003, PBT022, and PBT707 cells were infected with OVmIL15/21 (MOI = 1, 24 h) and analyzed by Western blot. β-actin (ACTB) is a loading control. Representative blots from $n = 2$ independent experiments. **C** mIL15 and mIL21 expression on GBM cells (PBT003, PBT022) ± OVmIL15/21 infection (MOI = 1, 24 h). Scale bar: 20 μm. Representative images from $n = 2$ independent experiments. **D** Quantification of vaccinia-, mIL15-, and mIL21-positive GBM cells (PBT003, PBT022) by flow cytometry. **E** GBM cell viability (PBT003, PBT022) after OVmIL15/21 infection, measured by luciferase assay. $n = 3$ technical replicates. **F** BiCAR-T cell expansion was measured by CD3⁺ cell counts after co-culture with GBM cells (PBT003, PBT022) infected with OVDual+OVmIL15/21 or OVDual+OVGFP

(MOI = 0.5). $n = 4$ cell culture replicates. Data are presented as mean ± SD and were analyzed by two-way ANOVA with Tukey's multiple comparisons test. *$p$ (left for PBT003) = 0.0115, *$p$ (right for PBT022) = 0.0148, ****$p < 0.0001$. **G** IFNγ production was measured in conditioned medium from PBT003 cells ± OVDual (MOI = 0.5) and OVmIL15/21 (MOI = 0.5 or 1), followed by BiCAR-T treatment. Untreated cells served as a control. $n = 3$ cell culture replicates. Data are presented as mean ± SD and were analyzed by one-way ANOVA with Tukey's multiple comparisons test. ***$p = 0.0004$, ****$p < 0.0001$. **H** BiCAR-T cytotoxicity against GBM cells (PBT003, PBT022) ± indicated OV combinations were measured by luciferase assay at different E:T ratios for 12, 24, or 48 h. $n = 3$ cell culture replicates. Data are presented as mean ± SD and analyzed by two-way ANOVA with Tukey's multiply comparisons test. *$p$ (PBT003-12h) = 0.0322, ***$p$ (PBT003-24h and PBT003-48h) = 0.0002, *$p$ (PBT022-12h) = 0.0147, ****$p$ (PBT022-24h) < 0.0001, and ns means no significant difference. Source data are provided in the Source Data file.

reporter-bearing GBM cells (PBT003-luc) were intracranially injected into the frontal lobe of NSG mice on day -25. On day -18 and -9, tumor-bearing mice were treated with OVDual plus OVmIL15/21. Mice treated with OVDual plus OVGFP or received no treatment were included as a control. Two days after OV treatment, mice received BiCAR-T cells or vehicle control (Fig. 6A). Bioluminescence imaging revealed that mice received OVDual plus OVmIL15/21 together with BiCAR-T cells had significantly reduced tumor burden and prolonged survival, compared to mice treated with OVDual plus OVGFP together with BiCAR-T cells, indicating that OVmIL15/21 can enhance the anti-tumor efficacy of the combination therapy of OVDual and BiCAR-T cells in vivo (Fig. 6B–D). The enhancing effect of OVmIL15/21 was dependent on the presence of BiCAR-T cells in the treatment, because we did not detect statistically reduced tumor progression when comparing the tumor size in mice treated with OVDual plus OVmIL15/21 vs that in mice treated with OVDual plus OVGFP without BiCAR-T cells (Fig. 6B, C). Moreover, when mice were treated with control T cells plus OVDual and OVmIL15/21 vs control T cells plus OVDual and OVGFP, there is no statistically significant difference in tumor progression and tumor-bearing mouse survival (Figure. S11A–D). We also detected substantially reduced tumor progression and prolonged survival in mice treated with BiCAR-T plus OVDual and OVmIL15/21, compared to mice treated with control T plus OVDual and OVmIL15/21 (Figure. S11A–D), indicating that the OVmIL15/21-boosted anti-tumor effect is dependent on BiCAR on T cells. These results together indicate that the combination of OVDual and BiCAR-T cells is a potent anti-tumor strategy for GBM and that OVmIL15/21 further enhances the efficacy of this combination therapy.

## BiCAR-NK cells exhibit potent anti-tumor effect when combined with OVDual and OVmIL15/21 in vitro

In addition to CAR-T cells, CAR-NK cells have emerged as another type of cell-based cancer immunotherapy with less toxicity, including cytokine release syndrome and neurotoxicity[57]. To test if our OVDual and BiCAR approach can also be applied to NK cell-based therapy, we intended to generate BiCAR-NK cells. However, the efficiency of lentiviral transduction in primary NK cells is low and multiple rounds of transduction are often needed[57]. NK cells derived from CAR-engineered human pluripotent stem cells (hPSC) can overcome this challenge and reach high transduction efficiency of CAR because PSC colonies derived from single transduced PSCs can be selected[50]. Moreover, because hPSCs have the ability to self-renew and expand without limitation[58], hPSC-derived NK cells can be massively produced and used as off-the-shelf cell therapies for many patients. To make BiCAR-NK cells, we first generated BiCAR-engineered human H9 embryonic stem cells (ESCs) by lentiviral transduction with the BiCAR vector. Single PSC colonies were picked after transduction and verified for the expression of BiCAR by flow cytometry (Fig. 7A). We then differentiated the BiCAR-transduced ESCs into NK cells following our published protocol[49] adapted from the spin-embryoid body (EB) method[59] (Fig. 7B). The expression of the NK cell marker CD56 and the

hematopoietic cell marker CD45 were detected by flow cytometry (Fig. 7C). The surface expression of BiCAR on NK cells derived from BiCAR-transduced ESCs was confirmed by flow cytometry using antibodies against the CD34 tag and the G4S linker, whereas no BiCAR expression was detected on NK cells derived from control ESCs (Fig. 7D). To assess BiCAR-NK functionality, we evaluated the cytotoxicity of BiCAR-NK cells against OVDual-infected GBM cells. A significantly enhanced killing efficacy was detected when BiCAR-NK cells were cocultured with OVDual-infected GBM cells, compared to that when control NK cells were cocultured with OVDual-infected GBM cells or when no NK cells were included but OVDual treatment alone (Fig. 7E). To demonstrate the functionality of both CD19 CAR and EGFRvIII CAR components in BiCAR-NK cells, we evaluated their cytotoxicity against GBM cells infected with either OVCD19t or OVEGFRvIIIt antigen. BiCAR-NK cells exhibited enhanced killing activity when cocultured with OVCD19t- or OVEGFRvIIIt-infected GBM cells, compared to control NK cells (Figure. S12A and B), indicating that both CD19 CAR and EGFRvIII CAR are functional in BiCAR-NK cells. Flow cytometric analysis revealed that BiCAR-NK cells exhibited markedly elevated activation when cocultured with OVDual-infected GBM cells (BiCAR-NK+OVDual) as revealed by increased expression of activation markers CD69, CD107a, and IFNγ, compared to BiCAR-NK cells cocultured with OVGFP-infected GBM cells (BiCAR-NK + OVGFP) or untreated GBM cells (BiCAR-NK) (Fig. 7F). Increased expression of CD69, CD107a, and IFNγ was also detected in BiCAR-NK cells cocultured with OVCD19t or OVEGFRvIII-infected GBM cells compared to BiCAR-NK + OVGFP or BiCAR-NK (Figure. S12C and D). These findings indicate that OVDual infection enhances BiCAR-NK activation and cytolytic capacity.

To assess the effect of mIL15/21 on NK cell expansion and cytotoxicity, GBM cells were first transduced with LVmIL15/21 and the transduced GBM cells were cocultured with BiCAR-NK cells derived from H9 ESCs. LVmIL15/21-transduced GBM cells significantly improved NK cell expansion compared to control GBM cells (Fig. 7G). Moreover, NK cells demonstrated significantly enhanced cytotoxic activity against LVmIL15/21-transduced GBM cells compared to control GBM cells (Fig. 7H). Next, GBM cells were infected with OVmIL15/21 and the transduced GBM cells were cocultured with BiCAR-NK cells. Infection of GBM cells with OVmIL15/21 resulted in a substantial increase in the expansion of cocultured NK cells compared to infection with GBM cells with the OVGFP control (Fig. 7I). NK cells also exhibited significantly enhanced cytotoxic activity against GBM cells infected with OVmIL15/21 relative to those infected with OVGFP (Fig. 7J). These results indicate that OVmIL15/21 enhances NK cell expansion and cytotoxicity.

## Off-the-Shelf BiCAR-NK cells exhibit potent anti-tumor effect when combined with OVDual and OVmIL15/21 in a xenograft GBM mouse model

To determine the effect of OVmIL15/21 on BiCAR-NK cells in vivo, we transplanted RFP-labeled GBM cells (PBT003-RFP) into the frontal lobe

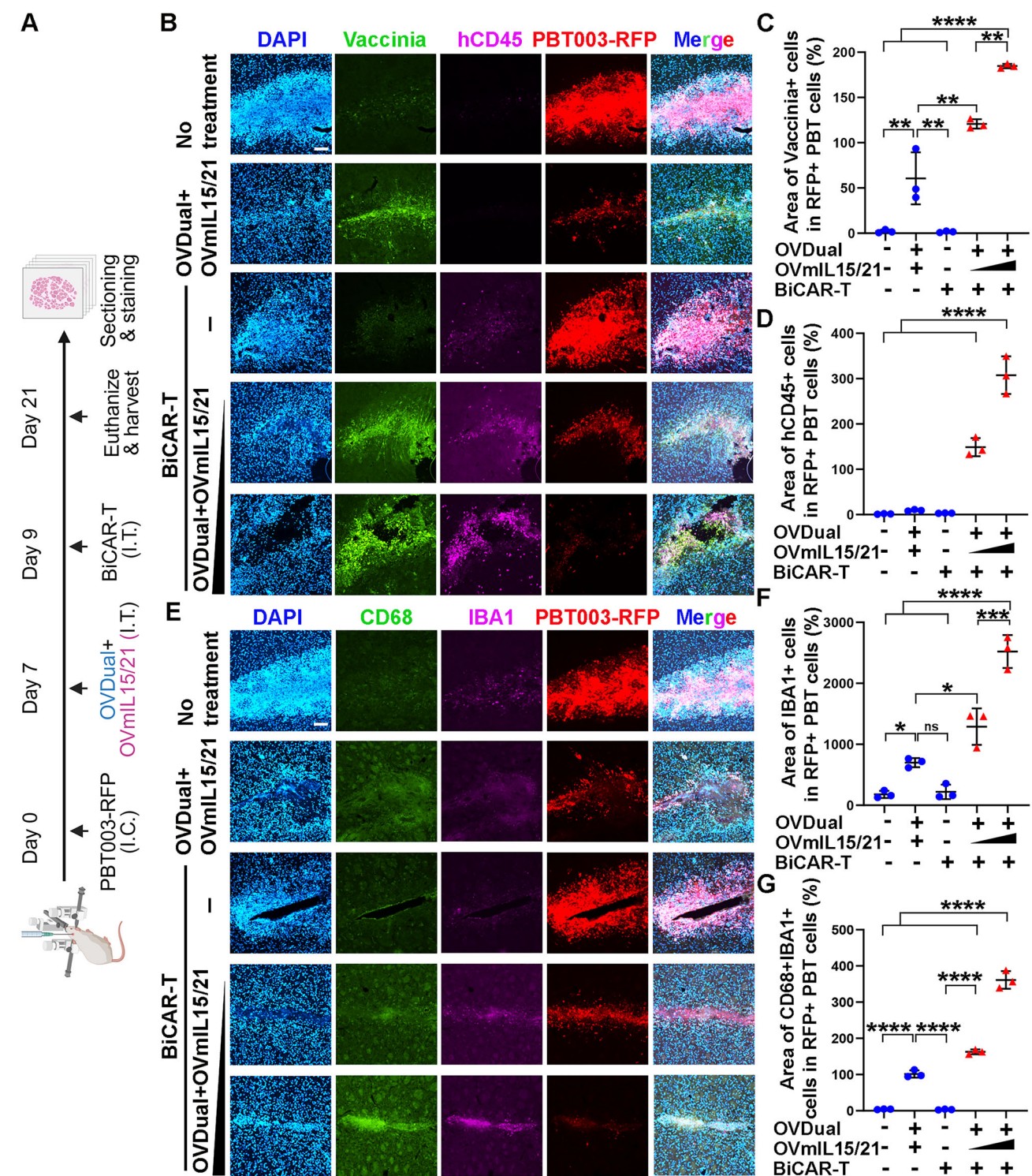

of NSG mice to establish tumors. 7 days after GBM cell transplantation, we administered OVDual ($3 \times 10^6$ PFU) combined with OVmIL15/21 or OVGFP ($3 \times 10^6$ PFU) into transplanted mice, followed by BiCAR-NK treatment 2 days after OV administration (Figure. S13A). Brain sections were analyzed 2 weeks or 4 weeks after OV administration using an anti−human CD56 (hCD56) antibody. We were able to detect hCD56+ NK cells at both 2-week and 4-week time points in BiCAR-NK-engrafted mice (Figure. S13B and C). Moreover, treatment with OVmIL15/21 + OVDual resulted in a higher percentage of hCD56-positive area within RFP-positive tumor tissues, compared to treatment with OVGFP + OVDual (Figure. S13B and C). These results indicate that mIL15/IL21

delivered by OVmIL15/21 can enhance NK cell persistence and/or infiltration into tumor tissue.

To evaluate the preclinical anti-tumor efficacy of BiCAR-NK cells in combination with OVDual in vivo, we tested the effect of this combination in a xenograft GBM mouse model. We included OVmIL15/21 in the combination because OVmIL15/21 enhanced the expansion and cytolytic capacity of BiCAR-NK cells in vitro (Fig. 7H, I). Luciferase-bearing GBM cells (PBT003-luc) were transplanted into the frontal lobe of NSG mice on day -25. The transplanted mice were treated with OV (OVDual plus OVGFP or OVDual plus OVmIL15/21) intratumorally on day -18 and day -9. Two days after OV treatment, these mice

**Fig. 5 | OV encoding mIL15/21 modulates the tumor microenvironment in a xenograft GBM mouse model. A** A schematic showing the experimental procedure (generated using BioRender). **B** Representative images of tumor-bearing mouse brains showing OV (vaccinia[+]), BiCAR-T cells (hCD45[+]), and tumor mass (RFP[+] GBM cells, PBT003-RFP) after the indicated treatments. Scale bar, 100 μm. **C** Quantification of OV-targeted PBT cell area within RFP[+] tumor regions. $n = 3$ mice per group, data from three brain slices were averaged to generate a single biological replicate. Data are presented as mean ± SD and were analyzed by one-way ANOVA followed by Tukey's multiple comparison test. **$p$ (OVDual+OVmIL15/21 vs no treatment) = 0.0020, **$p$ (OVDual+OVmIL15/21 vs BiCAR-T) = 0.0019, **$p$ (low-dose OVDual+OVmIL15/21+BiCAR-T vs OVDual+OVmIL15/21) = 0.0017, **$p$ (high-dose OVDual+OVmIL15/21+BiCAR-T vs low-dose OVDual+OVmIL15/21 + BiCAR-T) = 0.0010, and ****$p < 0.0001$. **D** Quantification of hCD45[+] BiCAR-T cell area within RFP[+] tumor regions. $n = 3$ mice per group, data from three brain slices were averaged to generate a single biological replicate. Data are presented as mean ± SD

and were analyzed by one-way ANOVA followed by Tukey's multiple comparison test. ****$p < 0.0001$. **E** Representative images of tumor-bearing mouse brains showing IBA1[+] and CD68[+] microglia and RFP[+] GBM cells after indicated treatments. Scale bar, 100 μm. **F** Quantification of IBA1[+] area within RFP[+] tumor regions. $n = 3$ mice per group, data from three brain slices were averaged to generate a single biological replicate. Data are presented as mean ± SD and were analyzed by one-way ANOVA with Tukey's multiple comparison test. *$p$ (OVDual+OVmIL15/21 vs no treatment) = 0.0480, *$p$ (low-dose OVDual+OVmIL15/21+BiCAR-T vs OVDual +OVmIL15/21) = 0.0244, ***$p$ (high-dose OVDual+OVmIL15/21+BiCAR-T vs low-dose OVDual+OVmIL15/21+BiCAR-T) = 0.0001, and ****$p < 0.0001$. **G** Quantification of CD68[+] and IBA1[+] area within RFP[+] tumor regions. $n = 3$ mice per group, data from three brain slices were averaged to generate a single biological replicate. Data are presented as mean ± SD and were analyzed by one-way ANOVA with Tukey's multiple comparison test. ****$p < 0.0001$. Source data are provided in the Source Data file.

received BiCAR-NK cell treatment on day -16 and -7 (Fig. 8A). Tumor progression was evaluated by weekly bioluminescence imaging. We first compared tumor progression in mice treated with the combination of BiCAR-NK, OVDual and OVmIL15/21 to that in mice with no treatment. The combination of BiCAR-NK with OVDual and OVmIL15/21 led to dramatically reduced tumor progression, compared to the no treatment control group (e.g. 93.1% vs 0% reduction in tumor luciferase activity relative to no-treatment control at week 2) (Fig. 8B, C), indicating the potent anti-tumor efficacy of this combination. Tumor progression in mice treated with BiCAR-NK plus OVDual and OVmIL15/21 was also reduced, compared to that in mice treated with BiCAR-NK plus OVDual and OVGFP (e.g. 93.1% vs 83.2% reduction in tumor luciferase activity relative to no-treatment control at week 2) (Fig. 8B, C), indicating that OVmIL15/21 can boost the anti-tumor effect of the combination of BiCAR-NK with OVDual. We also detected reduced tumor progression in mice treated with BiCAR-NK together with OVDual and OVmIL15/21, compared to that in mice treated with OVDual and OVmIL15/21 without BiCAR (e.g. 93.1% vs 68% reduction in tumor luciferase activity relative to no-treatment control at week 2) (Fig. 8B, C). This result indicates that BiCAR-NK is important in reducing tumor progression when administered in combination with OVDual and OVmIL15/21. Accordingly, we detected statistically significant improvement in the survival of tumor-bearing mice treated with BiCAR-NK together with OVDual and OVmIL15/21, compared to all the control groups (Fig. 8D). These results indicate that BiCAR-NK cells exert potent anti-tumor effect against GBM, when combined with OVDual and OVmIL15/21.

## Discussion

Our study using the combination of OVDual encoding the truncated EGFRvIII and CD19 and BiCAR against both antigens to target GBM provides a proof-of-principle that both GBM-specific and not-normally-GBM-expressed antigens can be delivered onto GBM tissues, and these antigens can be targeted simultaneously using BiCAR-T or NK cells to achieve potent anti-tumor effect. Moreover, mIL15/21 delivered to tumor cells by oncolytic virus can boost the anti-tumor effect of the combination of OVDual and BiCAR substantially.

Cancer immunotherapy using CAR-T cells has achieved significant success in patients with refractory B cell malignancies. However, there are challenges that have prevented the broad application of CAR-T therapy to other types of tumors. The outcome of CAR-T in patients with solid tumors, especially GBM, has not always met with success, presumably due to heterogeneous tumor antigen expression, antigen loss or escape following treatment with CAR-T cells targeting a single antigen, and the immunosuppressive tumor microenvironment[7,11,14,16,17]. Our study, by developing a multimodal approach, is designed to overcome these challenges of current CAR-T therapy to achieve robust and consistent effects of CAR-T treatment of GBM.

In this study, we developed an approach that strategically integrates BiCAR with OVDual into a combination therapy and demonstrated how this platform can be used to address critical limitations of current CAR-T therapy for solid tumors such as GBM, including tumor antigen heterogeneity, antigen escape, and the immunosuppressive tumor microenvironment. GBM is notorious for its heterogeneity including tumor antigen heterogeneity[60]. Moreover, antigen escape has been detected in clinical studies using CAR-T cells, especially in solid tumors[61]. While EGFRvIII is the most common variant of the EGFR observed in human tumors[18] and is expressed in about 30% to 50% of newly diagnosed GBM cases, in the first-in-human clinical trial, EGFR-vIII CAR-T cells reduced EGFRvIII-expression on tumor cells in GBM patients, but did not result in clear tumor regression due to the emergence of EGFRvIII-negative tumor cells[7], indicating the need to target EGFRvIII and other antigens expressed on GBM cells in order to develop effective EGFRvIII-based CAR-T therapies for GBM. In this study, we developed OVDual that encodes a truncated form of both EGFRvIII, a GBM-specific antigen, and CD19, an excellent target for CAR-T cells against hematological malignancies[21], and delivered both antigens onto GBM cells for CAR-targeting. We selected EGFRvIII as an antigen for GBM-targeting CAR based on the specific expression of EGFRvIII in GBM tumor cells but not normal cells, the safety profile of EGFRvIII CAR-T cells, and their ability to eradicate EGFRvIII-positive GBM cells in previous clinical trials[7,62]. We selected CD19 as an antigen because potent CD19 CAR has been developed and targeting CD19 with CAR-T cells is a clinically validated approach for treating hematological malignancies[22,23], supporting the future clinical development of our OVDual strategy. Moreover, it has been shown that CD19 can be introduced onto solid tumors as a CAR antigen for effective CAR targeting[63].

In parallel to OVDual, we developed bi-specific CAR-T cells to achieve dual-antigen targeting. The bi-specific CAR-T cells exhibited more potent tumor killing effect when combined with OVDual, compared to control T cells without CAR, in both 2D GBM cultures and 3D tumor spheroid models that can better mimic the 3D tumor tissues. The combination of OVDual with BiCAR-T cells led to better viral spreading, presumably due to the release of OV from CAR-T-lysed tumor cells, which can infect surrounding tumor cells and lead to more tumor antigen presentation on tumor tissues. Importantly, intratumoral delivery of BiCAR-T cells into mice bearing OVDual-infected GBM led to robust tumor regression and substantially prolonged survival of tumor-bearing mice.

Treatment with OVDual plus BiCAR-T enhanced the survival of tumor-bearing mice significantly compared to treatment with OVDual alone, BiCAR-T alone, or no treatment. Although the survival benefit of BiCAR-T cells combined with OVDual was modest over single EGFRvIII CAR-T cells combined with OVEGFRvIII in the xenograft NSG model, the results support further evaluation of this combination in clinical studies, because NSG mice are immunodeficient and do not

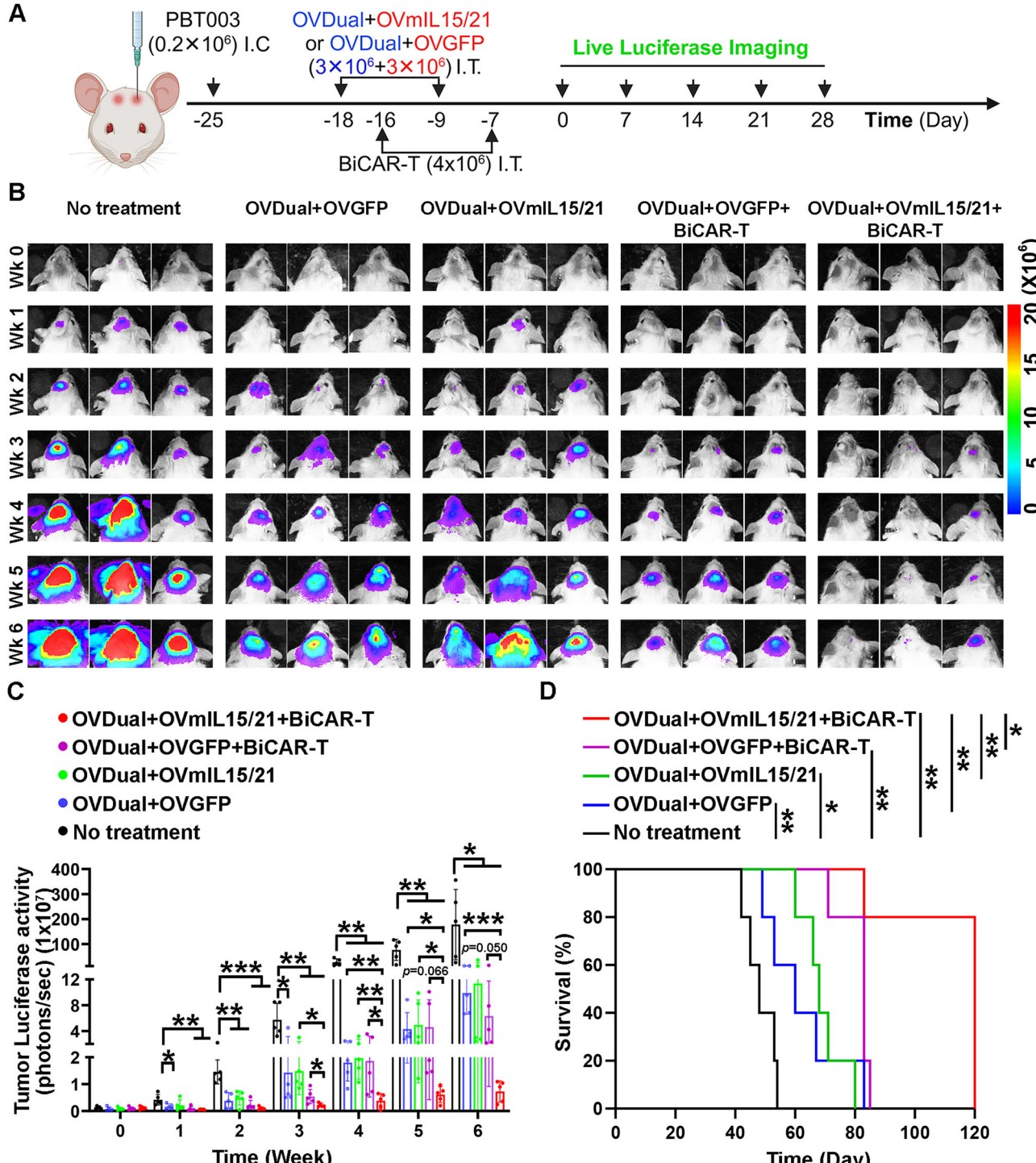

recapitulate the complex human GBM tumor microenvironment, which includes diverse immune population, cytokine network, and selective pressure that drive dynamic change in antigen expression. In GBM patients, antigen heterogeneity and antigen loss are major barriers that limit the effectiveness of CAR-T therapy using a single CAR[7,11,14–17]. Our dual-antigen-encoding OV provides an additional layer of protection by introducing a second, independent antigen, thereby reducing the likelihood that tumor cells will fully evade CAR-T targeting. When OVDual is paired with BiCAR-T cells capable of responding to either antigen, this strategy has the potential to prevent antigen escape and achieve more robust and durable anti-tumor response in the more complex human GBM microenvironment.

In addition to delivering tumor-associated antigens to GBM tissues, OV can also help address the challenge of the immunosuppressive tumor microenvironment for GBM therapeutic development because OV exhibits immune-stimulating effects and can turn the immunologically 'cold' tumor microenvironment to the 'hot'. Moreover, OV has been genetically engineered to express cytokines/chemokines to enhance the recruitment of tumor-targeting immune cells and promote antitumor efficacy[30]. In this study, we developed OV encoding both mIL15 and mIL21 and delivered OVmIL15/21 to GBM cells. IL15 and IL21 have been shown to promote immune cell survival, expansion, and functionality individually or together[40,44,53,64–67], however, whether mIL5 and mIL21 can be delivered together by OV to

**Fig. 6 | OVmIL15/21 enhances anti-tumor efficacy when used in combination with OVDual and BiCAR-T cells in a xenograft GBM model. A** A schematic illustration of the experimental procedure. I.C.: intracranial; I.T.: intratumoral (generated using BioRender). **B** Bioluminescence images of brain tumors in xenografted NSG mice. **C** Quantification of the bioluminescence intensity of tumors. $n = 5$ mice per group. Data are Mean ± SD and were analyzed by two-tailed unpaired t tests. Week1: $*p = 0.0325$, $**p = 0.0081$, and $**p = 0.0034$ (OVDual+OVGFP, OVDual+OVGFP+BiCAR-T, or OVDual+OVmIL15/21+BiCAR-T vs no treatment). Week2: $**p = 0.0020$, $**p = 0.0031$, $***p = 0.0004$, $***p = 0.0001$ (OVDual+OVGFP, OVDual+OVmIL15/21, OVDual+OVGFP+BiCAR-T, or OVDual+OVmIL15/21+BiCAR-T vs no treatment). Week3: $*p = 0.0137$, $**p = 0.0075$, $**p = 0.0018$, and $**p = 0.0012$ (OVDual+OVGFP, OVDual+OVmIL15/21, OVDual+OVGFP+BiCAR-T, or OVDual+OVmIL15/21+BiCAR-T vs no treatment); $*p = 0.0123$, $*p = 0.0332$ (OVDual+OVmIL15/21+BiCAR-T vs OVDual+OVmIL15/21 or OVDual+OVGFP+BiCAR-T). Week4: $**p = 0.0024$, $**p = 0.0025$, $**p = 0.0025$, $**p = 0.0018$ (OVDual+OVGFP, OVDual+OVmIL15/21, OVDual+OVGFP+BiCAR-T, or OVDual+OVmIL15/21+BiCAR-T vs no treatment); $**p = 0.0025$, $**p = 0.0027$, $*p = 0.0417$ (OVDual+OVmIL15/21+BiCAR-T vs OVDual+OVGFP, OVDual+OVmIL15/21, or OVDual+OVGFP+BiCAR-T). Week5: $**p = 0.0050$, $**p = 0.0053$, $**p = 0.0052$, $**p = 0.0037$ (OVDual+OVGFP, OVDual+OVmIL15/21, OVDual+OVGFP+BiCAR-T, or OVDual+OVmIL15/21+BiCAR-T vs no treatment); $*p = 0.0119$, $*p = 0.0382$ (OVDual+OVmIL15/21+BiCAR-T vs OVDual+OVGFP or OVDual+OVmIL15/21). Week6: $*p = 0.0282$, $*p = 0.0298$, $*p = 0.0259$, $*p = 0.0225$ (OVDual+OVGFP, OVDual+OVmIL15/21, OVGFP+BiCAR-T, or OVDual+OVmIL15/21+BiCAR-T vs no treatment); $***p = 0.0003$ (OVDual+OVmIL15/21+BiCAR-T vs OVDual+OVGFP). **D** The survival of GBM cell-bearing mice. All remaining mice were euthanized on day 120. $n = 5$ mice per group. Data were analyzed by log-rank test. $**p = 0.0018, 0.0415, 0.0018, 0.0018$ (OVDual+OVGFP, OVDual+OVmIL15/21, OVDual+OVGFP+BiCAR-T, or OVDual+OVmIL15/21+BiCAR-T vs no treatment); $**p = 0.0018, 0.0044, 0.0128$ (OVDual+OVmIL15/21+BiCAR-T vs OVDual+OVGFP, OVDual+OVmIL15/21, or OVDual+OVGFP+BiCAR-T). Source data are provided in the Source Data file.

activate the tumor microenvironment remains to be tested. In this study, we delivered mIL15 and mIL21 to GBM cells by OV and showed that OVmIL15/21 enhanced CAR-T and CAR-NK cell expansion and cytotoxicity in vitro and promoted the anti-tumor effect of CAR-T and CAR-NK cells in vivo. The strong boosting effect of OVmIL15/21 leads to a more potent anti-tumor platform by combining BiCAR and OVDual with OVmIL15/21.

The combination of BiCAR-T or BiCAR-NK cells with OVDual and OVmIL15/21 could have broader applicability, not only for GBM but also for many other tumor types. The specific BiCAR-T cells with tandem EGFRvIII CAR and CD19 CAR that we developed in this study can be used for multiple tumor types that have demonstrated heterogenous EGFRvIII expression, including not only GBM[68], but also medulloblastoma, breast cancer, and ovarian cancer[69], as well as CD19-positive acute lymphoblastic leukemia, chronic lymphocytic leukemia, and B cell lymphomas[70–73]. For OVDual, in addition to the EGFRvIII and CD19 antigen pair, we can also deliver other antigen pairs including antigens that are not normally expressed on specific tumor tissues to those tumor tissues by OV to expand the targetable tumor antigen list for specific tumor tissues.

The use of hPSC-derived NK cells represents a promising advancement for developing cell-based cancer immunotherapies. hPSCs are self-renewable and can expand extensively, therefore can provide unlimited cell source for generating NK cells. Moreover, hPSCs can be easily engineered to express specific CARs and can be clonally purified before NK differentiation, providing a solution to overcome the challenge of low CAR transduction in primary NK cells. Furthermore, hPSC-NK cells can be developed into "off-the-shelf" products[45], which can eliminate patient-specific manufacturing delays and cost. In this study, we demonstrated that combining hPSC-derived CAR-NK cells with oncolytic virotherapy enhanced anti-tumor responses in GBM models substantially, therefore offering a promising therapeutic option for GBM, for which there is a high unmet medical need[74,75]. This approach can also be applied to other types of hard-to-treat tumors.

In summary, we have demonstrated that OV can effectively deliver both GBM-specific and not-normally-GBM-expressed tumor antigens to GBM. BiCAR-T cells can effectively eradicate these OV-infected GBM cells. OV encoding mIL15/21 can further boost the anti-tumor effect of BiCAR in combination with OVDual by enhancing immune cell expansion/persistence and functionality.

These findings establish preclinical proof of concept for a multimodal targeting strategy in GBM, demonstrating the feasibility of combining dual-antigen-encoding oncolytic viruses with engineered BiCAR immune cells. The platform provides a modular framework for investigating combinatorial targeting approaches. These findings provide a foundation for future clinical studies that may pave the way for a promising approach in the treatment of GBM. The multimodal

therapeutic platform that we established in this study can be applicable not only to GBM but also other types of intractable cancers.

## Methods
### Ethics statement
Tumor growth in mice was monitored according to the protocol approved by the City of Hope Institutional Animal Care and Use Committee under protocol number 24025. Although exact tumor volume was not measured due to the intracranial location of the tumor, humane endpoints were defined in the protocol, including moribund condition, neurological deficits, or significant weight loss (≥20%). No animals exceeded these endpoints during the study.

Specimens without identifiers from leftover surgical samples and healthy blood donors were used in this study. The protocol was reviewed by the City of Hope Institutional Review Board (IRB) and was determined not to involve human subjects research.

### Animals
NOD/SCID/IL-2rg (NSG) mice, 8-20 weeks of age, were obtained from The Jackson Laboratory, bred at the Animal Resources Center of City of Hope, and used in all experiments described in this study. All animal procedures were approved by the City of Hope Institutional Animal Care and Use Committee under protocol number 24025. Mice were housed under controlled conditions (temperature 20–24 °C, humidity 30–70%) with a 12 h light/dark cycle. Animals were euthanized upon reaching predefined humane endpoints, including moribund condition, neurological deficits, or significant weight loss (≥20%). Sex was not considered as a biological variable in this study. The experiments were designed to evaluate tumor growth and therapeutic response in an immunodeficient xenograft model, and no sex-specific effects have been reported for these endpoints in NSG mice. Accordingly, animals were not stratified by sex for allocation or analysis.

### Cell lines
H9 ESCs (derived from female human blastocysts) were purchased from WiCell (Cat# WA09). PBMCs were isolated from an adult male. H9 ESCs were maintained on Matrigel (1:100 in DMEM/F12) (Fisher, Cat# CB 40230)-coated plate with mTeSR-plus medium (StemCell Technologies, Cat# 100–0276). HEK293T cells were obtained from the American Type Culture Collection (ATCC, Cat# CRL-11268) and maintained in DMEM medium supplemented with 10% fetal bovine serum and antibiotic. GBM cells were derived from patients with newly diagnosed grade IV GBM and maintained in spheres in DMEM/F12 medium supplemented with 2 mM L-glutamine, 27.4 mM HEPES, 1 × B27, 20 ng/ml epidermal growth factor, 20 ng/ml fibroblast growth factor, and 5 µg/ml heparin as described[76–78]. All cultures were mycoplasma free as confirmed using a MycoAlert PLUS Mycoplasma Detection Kit (Lonza, LT07-318). African green

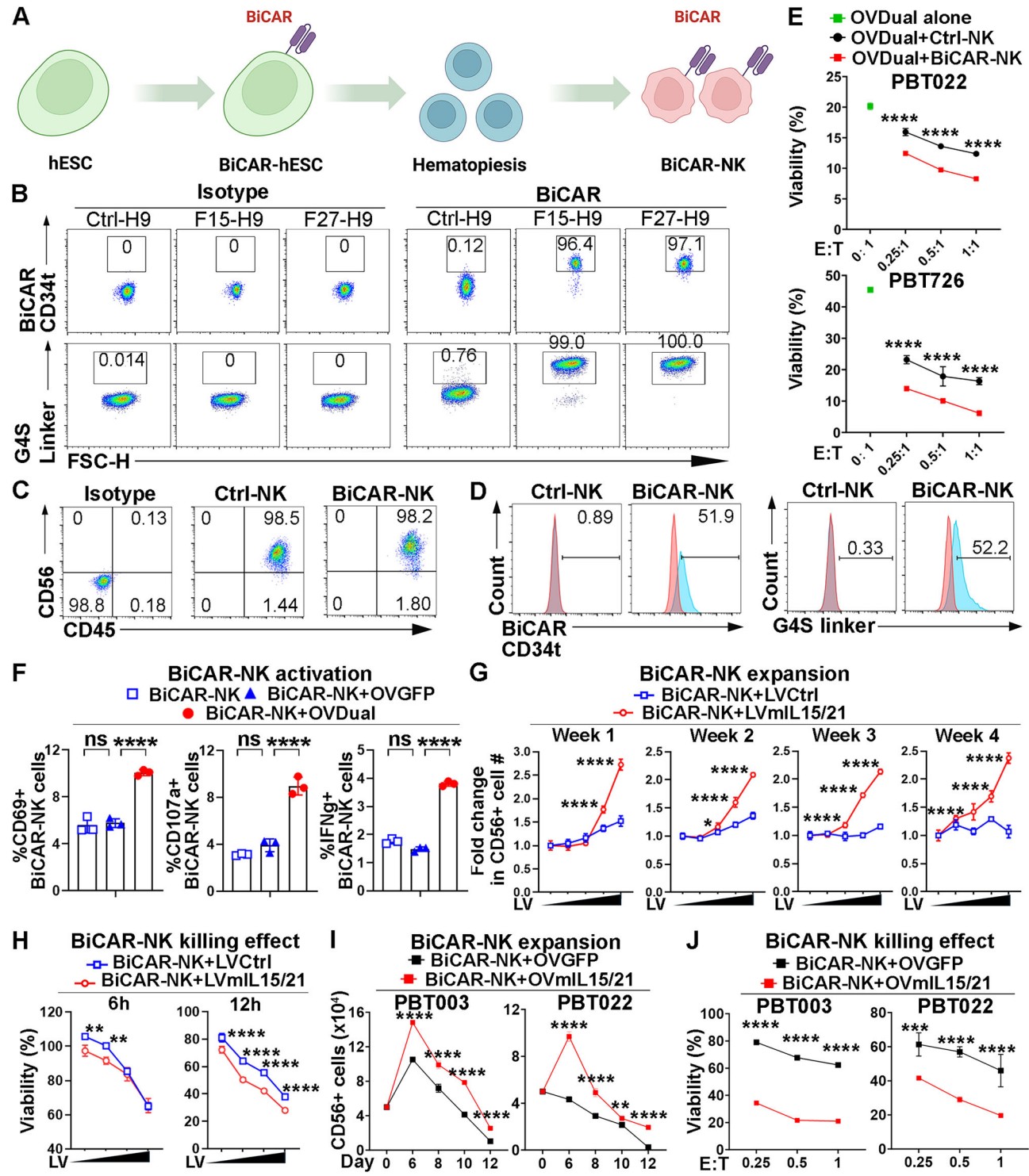

monkey kidney fibroblasts (CV-1) were used to amplify and titrate oncolytic virus.

### Generation of recombinant chimeric orthopoxvirus

The recombinant oncolytic virus encoding membrane bound human IL15 and IL-21 (CF17-mIL15-IL21) was constructed using homologous recombination technique, and then CRISPR/Cas9 technology was used to assist with the selection of the recombinant virus as previously described[79]. Briefly, the cDNA for mIL15 and mIL21 connected by P2A was PCR amplified and cloned into a shuttle plasmid to generate pSC32 (J2R-shuttle plasmid) where the mIL15-P2A-mIL21-expression cassette (cDNA driven by a viral

synthetic early/late promoter pSE) was flanked by ~500 bp of CF17 sequence from around the J2R locus.

For using CRISPR/Cas9 in the selection of desired recombinant viruses, a J2R specific oligo was designed and purchased from IDT. The oligo (5′-GTTATAGTAGCCGCACTCG-3′) was used to synthesize single guide RNA (sgRNA) using the EnGen sgRNA Synthesis Kit (NEB, Cat# E3322S) following the manufacturer's instruction. Next, ~70% confluent HEK293 cells in a 24 well plate were infected with CF17 virus at an MOI of 0.1. Two h after infection, medium was changed to fresh DMEM medium supplemented with 10% FBS, and the cells were transfected with ribonucleoprotein complex of sgRNA and Cas9 together with the template for homologous recombination i.e., the pSC32 plasmid

**Fig. 7 | OVmIL15/21 enhances the survival and cytotoxicity of BiCAR-NK cells in vitro. A** A schematic showing BiCAR transduction of hESC cells and the differentiation of BiCAR-hESCs into NK cells (generated using BioRender). **B** Representative flow cytometry plots of BiCAR expression in hESC clones. **C** Representative flow cytometry plots of CD45 and CD56 expression on NK cells. **D** Representative flow cytometry plots of BiCAR expression on Ctrl-NK and BiCAR-NK cells. **E** Enhanced cytotoxicity of BiCAR-NK cells against OVDual-infected GBM cells, measured by luciferase assay. $n = 3$ cell culture replicates. Data are presented as mean ± SD and analyzed by two-way ANOVA with Tukey's multiple comparisons test. ****$p < 0.0001$. **F** Enhanced activation of BiCAR-NK cells co-cultured with GBM cells infected with OVDual, compared to OVGFP-infected or untreated GBM cells. $n = 3$ cell culture replicates. Data are presented as mean ± SD and analyzed by one-way ANOVA with Tukey's multiple comparisons test. ****$p < 0.0001$ and ns means no significant difference. **G** Enhanced expansion of CD56+ NK cells co-cultured with GBM cells infected with LVCtrl or LVmIL15/21 at the indicated dose. $n = 3$ cell culture replicates. Data are presented as mean ± SD and analyzed by two-way ANOVA with Tukey's multiple comparisons test. *$p = 0.0312$ and ****$p < 0.0001$. **H** Enhanced cytotoxicity of BiCAR-NK cells against GBM cells transduced with LVmIL15/21 measured by luciferase assay. $n = 3$ cell culture replicates. Data are presented as mean ± SD and analyzed by two-way ANOVA with Tukey's multiple comparisons test. **$p$ (6h-left) = 0.0018, **$p$ (6h-right) = 0.0013, and ****$p < 0.0001$. **I, J** OVmIL15/21-infected GBM cells (PBT003, PBT022) enhanced BiCAR-NK cell expansion and cytotoxicity. $n = 3$ cell culture replicates. Data are presented as mean ± SD and analyzed by two-way ANOVA with Tukey's multiple comparisons test. **$p = 0.0026$ and ****$p < 0.0001$ for (**I**), ***$p = 0.0004$ and ****$p < 0.0001$ for (**J**). Source data are provided in the Source Data file.

containing mIL15-P2A-mIL21 expression cassette. Lipofectamine CRISPRMAX transfection reagent (Thermo Fisher, Cat# CMAX00008) was used for transfection; 1250 ng of Cas9 and 240 ng sgRNA was used per well. The following day, cell lysate was collected and subjected to 3 rounds of freeze-thaw cycles. The lysates were then used to infect CV1 cells in 6-well plates and methylcellulose overlay was applied. Two days after incubation, plaques were picked, and PCR verified. Two more rounds of plaque purification were performed in 6-well plates using methylcellulose overlay. A single purified plaque was then used to produce seed stock in a 100 mm dish of A549 cells. The seed stock was then used to further amplify the virus, and the virus was purified first on a sucrose cushion followed by further purification on sucrose gradient using high-speed centrifugation. Sanger sequencing was used to verify the transgene sequence in the recombinant virus.

The CF17-CD19t-EGFRvIIIt virus was constructed by inserting the cDNA for truncated CD19 (CD19t) and EGFRvIII (EGFRvIIIt) into the J2R locus and the F14.5L locus in the CF17 virus, respectively. The selection and purification of the virus was done as described above.

### Generation of CAR-transduced H9 ESCs
H9 ESCs (female, WiCell, Cat# WA09) were transduced with lentivirus encoding BiCAR with truncated CD34 as a tag. The transduced cells were cultured for at least two passages and then subjected to cell sorting by flow cytometry. Clonal CAR-positive cells were selected and expanded in mTeSR plus medium on Matrigel-coated plates and banked for subsequent differentiation. Two of the clones were selected randomly and confirmed for BiCAR expression. These cells were used for NK differentiation and further experiments.

### Differentiation of hESCs into NKs
NK cells were generated from human pluripotent stem cells (hPSCs) using the STEMdiff™ and StemSpan™ NK Cell kits (StemCell Technologies, Cat# 100-0170 and 09960), following the manufacturer's instructions with modifications to optimize the process. Briefly, spin embryoid bodies (EBs) were generated to produce CD34+ hematopoietic progenitor cells (HPCs). The CD34+ HPCs were differentiated into NK cells for two weeks and then transition into NK maturation culture for an additional two weeks. Following maturation, the NK cells were cocultured with irradiated K562-mbIL-21 artificial antigen-presenting cells (aAPCs)[42] to facilitate expansion over four weeks.

### Generation of CAR-T cells
Fresh PBMCs were stimulated with Dynabeads Human T-Expander CD3/CD28 (Fisher Scientific, Cat# 11131D) at a ratio of 1:3 (cell:bead) in RPMI 1640 medium (Gibco, Cat# 11875085) containing 10% FBS, 25 U/mL rhIL-2. Cells were transduced with lentivirus to express BiCAR with 5 μg/ml polybrene following a published procedure[80]. Cultures were then maintained under the same medium and cytokine conditions. Fresh cytokines were supplied every other day. On day 9 after transduction, the CD3/CD28

Dynabeads were removed from the culture using the DynaMag-50 magnet (Fisher Scientific, Cat# 12321D). The cells were expanded in culture until harvest at day 17 or as indicated.

The PBMC-derived CAR-T cells were sorted by flow cytometry. The sorted CAR-T cells were cultured on $50 \times 10^6$ γ-irradiated (35 Gy) PBMCs and $10 \times 10^6$ γ-irradiated K562-mIL21 cells (80 Gy) in 50 mL RPMI 1640 medium containing 10% FBS, 20 ng/mL anti-CD3 (Miltenyi Biotec, Cat# 130-093-377), 50 U/mL rhIL-2 and 10 ng/mL rhIL-7. The cultures were maintained for 14 days with half-volume medium changes every 2 days.

### Human subject research
Specimens without identifiers from leftover surgical tissues were used in this study. The information was evaluated and determined not to involve human subject research by the City of Hope Institutional Review Board.

### Plasmid DNAs
Membrane-bound human IL15 and IL21 were cloned into AAVS1-TRE3G lentiviral vector (Addgene, plasmid# 52343). The plasmid of membrane-bound IL-15 (mbIL15) was provided by Dr. Dario Campana from National University of Singapore. Briefly, assembly of the long signal peptide from CD8α, the mature IL-15 coding sequence, and the transmembrane domain of CD8α led to the generation of membrane-bound IL15 (mIL15)[34]. The design of Membrane-bound IL-21 followed a similar strategy. The cDNA encoding human IL-21 was PCR amplified from HEK293T cells, and the mature IL-21 sequence was fused to the signal peptide and transmembrane domain of CD8α through overlapping extension PCR to create a membrane-bound form of IL-21 (mbIL21). The mbIL15 and mbIL21 cDNAs were linked via the self-cleaving 2 A peptide (P2A) to ensure equimolar expression of both membrane-bound cytokines. The final recombinant construct encoded the fusion sequence of CD8α signal peptide/mature IL-15/CD8α transmembrane domain-P2A-CD8α signal peptide/mature IL-21/CD8α transmembrane domain. The construct was verified by DNA sequencing before subsequent expression analyses.

The EGFRvIII CAR was provided by Dr. Carl June from University of Pennsylvania [20] and modified to make human Ubc-EGFRvIII CAR-CD19t lentiviral vector. The CD19 CAR was described previously[81,82]. To construct the BiCAR (EGFRvIII CAR-CD19 CAR) lentiviral construct, we followed the bi-specific CAR design strategy from a published work[83] to construct our tandem EGFRvIII-CAR-CD19-CAR. Specifically, the EGFRvIII-CAR was positioned upstream of the CD19-CAR, and the two were linked using a (G4S)4 linker. The EGFRvIII-CAR-(G4S)4-CD19-CAR DNA fragment was assembled by fusion PCR and cloned into the Ubc-EGFRvIII-CAR-CD19t construct by replacing the EGFRvIII-CAR-CD19t fragment. To create the final BiCAR construct (Ubc-EGFRvIII-CAR-CD19-CAR-T2A-CD34t), a truncated version of CD34 (CD34t) amplified from CD34-bio-His (Addgene, plasmid #51647) was incorporated as a tag.

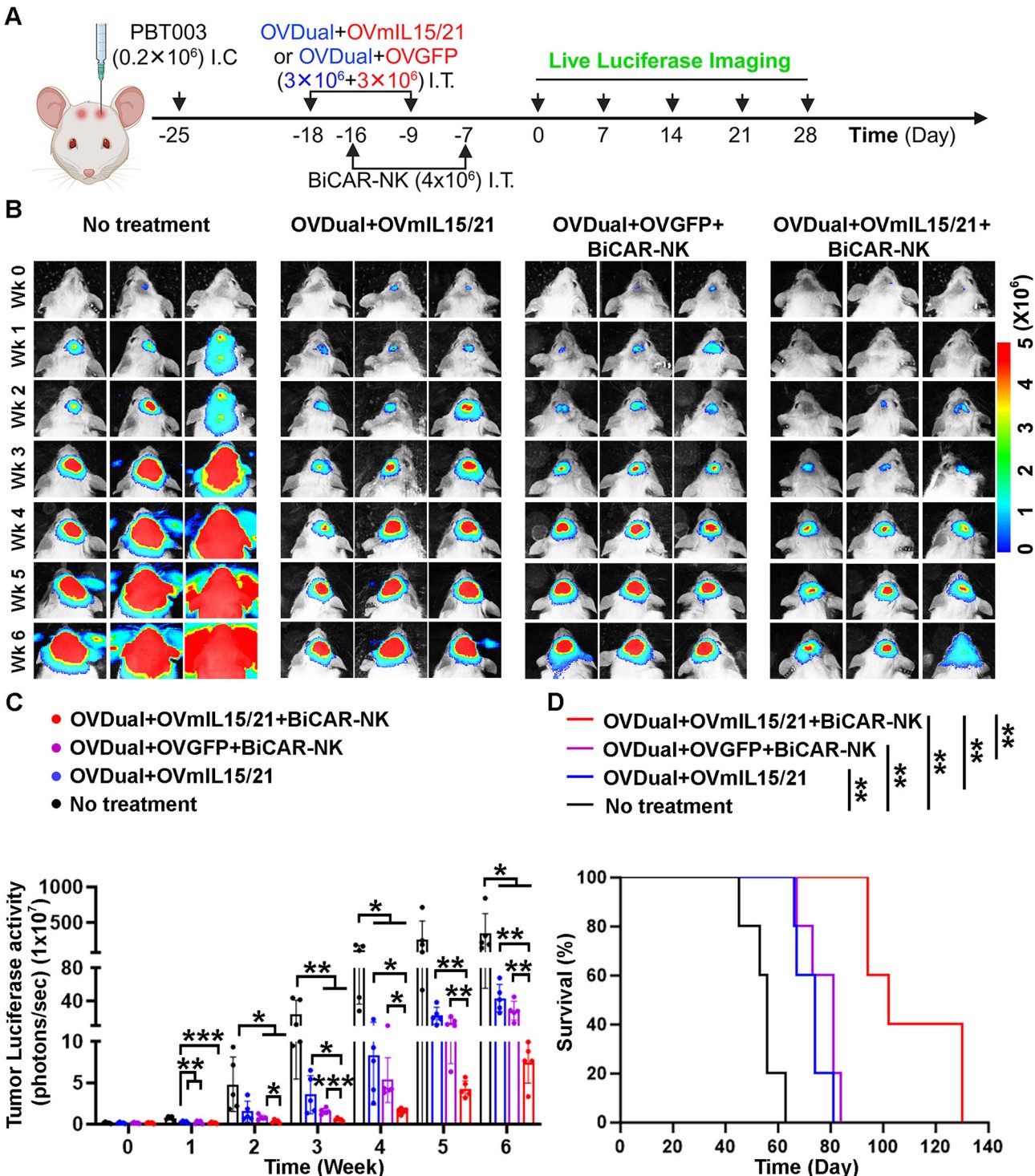

**Fig. 8 | OVmIL15/21 enhances the anti-tumor efficacy of the combination of BiCAR-NK cells and OVDual in a xenograft GBM mouse model. A** A schematic of experimental design. NSG mice bearing intracranial (I.C.) PBT003-luc tumors received intratumoral (I.T.) treatments (generated using BioRender). **B** Representative bioluminescence images of tumors. **C** Quantification of tumor bioluminescence intensity. *n* = 5 mice per group. Data are presented as mean ± SD and analyzed by two-tailed unpaired t tests. Week1: **$p$ = 0.0017, **$p$ = 0.0020, ***$p$ = 0.0003 (OVDual+OVmIL15/21, OVDual+OVGFP+BiCAR-NK, or OVDual+OVmIL15/21+BiCAR-NK vs no treatment). Week2: *$p$ = 0.0265, *$p$ = 0.0159 (OVDual+OVGFP+BiCAR-NK or OVDual+OVmIL15/21+BiCAR-NK vs no treatment); *$p$ = 0.0153 (OVDual+OVmIL15/21+BiCAR-NK vs OVDual+OVGFP+BiCAR-NK). Week3: **$p$ = 0.0068, **$p$ = 0.0057 (OVDual+OVGFP+BiCAR-NK or OVDual+OVmIL15/21+BiCAR-NK vs no treatment); *$p$ = 0.0199, ***$p$ = 0.0004 (OVDual

+OVmIL15/21+BiCAR-NK vs OVDual+OVmIL15/21 or OVDual+OVGFP+BiCAR-NK). Week4: *$p$ = 0.0146, *$p$ = 0.0114, *$p$ = 0.0104 (OVDual+OVmIL15/21, OVDual+OVGFP+BiCAR-NK, or OVDual+OVmIL15/21+BiCAR-NK vs no treatment); *$p$ = 0.0360, *$p$ = 0.0182 (OVDual+OVmIL15/21+BiCAR-NK vs OVDual+OVmIL15/21 or OVDual+OVGFP+ BiCAR-NK). Weeks 5: **$p$ = 0.0046 (OVDual+OVmIL15/21+BiCAR-NK vs OVDual+OVmIL15/21); **$p$ = 0.0096 (OVDual+OVmIL15/21+BiCAR-NK vs OVDual+OVGFP+BiCAR-NK). Weeks 6: **$p$ = 0.0014 (OVDual+OVmIL15/21+BiCAR-NK vs OVDual+OVmIL15/21); **$p$ = 0.0041 (OVDual+OVmIL15/21+BiCAR-NK vs OVDual+OVGFP+BiCAR-NK); *$p$ = 0.0484, 0.0312, 0.0401 (OVDual+OVmIL15/21, OVDual+OVGFP+BiCAR-NK, or OVDual+OVmIL15/21+BiCAR-NK vs no treatment). **D** The survival of tumor-bearing mice after indicated treatments. *n* = 5 mice per group. Data were analyzed by log-rank test. **$p$ = 0.0021. Source data are provided in the Source Data file.

## Lentiviral preparation and transduction

Lentiviruses were generated using 293 T cells as previously described[84]. GBM cells were transduced by incubating them with lentivirus in the presence of 4 µg/ml polybrene (AmericanBio, Cat# AB01643-00001) for 24 h. Transduced cells expressing the transgene were then selected using 2 µg/ml puromycin (Gibco, Cat# A1113803).

## Viral plaque assay

Cells were plated at confluence into 6-well plates in 2 mL growth medium and then infected with 0.01 MOI of each virus (OVDual or OVmIL15/21). Cells were harvested in triplicate for three consecutive days. CV-1 cells were infected with serial dilutions of samples treated with OV in 24-well plates.

## In vitro killing assays

For OV and CAR-T treatment and tumor killing assays, tumor cells were dissociated with Accutase (Gibco, Cat# A1110501) into single cells and seeded onto a round-bottom 96-well plate (FALCON, Cat# 34922025) and cultured for 2 h, then inoculated with OVDual at an MOI of 1 or at indicated MOI and infected for 4 h before adding BiCAR-T cells. The cells were cocultured and treated for 12, 24, or 48 h respectively. The cells were harvested and subjected to a luciferase reporter assay with the ONE-Glo™ Luciferase Assay System (Promega, Cat# E6120).

## Flow cytometry analysis

ESCs and GBM cells were dissociated with Accutase (Gibco, Cat# A1110501) to obtain single-cell suspension for staining. BiCAR-T cells derived from PBMCs were directly used for staining. DPBS containing 2% FBS (Sigma Aldrich, Cat# A7906) was used as the Flow Cytometry Staining buffer. For staining, cells were suspended and incubated with 50 µL of diluted antibodies for 30 min on ice. The detailed information of all the antibodies used was listed in Table S1. For vaccinia viral protein staining, cells were fixed and permeabilized using the BD Cytofix/Cytoperm Fixation/Permeabilization Solution Kit according to the manufacturer's protocol (BD Biosciences, Cat# 554714). Cells were then incubated with anti-vaccinia virus primary antibody (Abcam, Cat# ab35219) for 30 min on ice. Cells were washed twice and then stained with Alexa 488-conjugated anti-rabbit IgG (H + L) secondary antibody (Jackson ImmunoResearch, Cat# 711-545-152) for 30 min on ice. Cells were then washed twice, resuspended in FACS buffer, and analyzed on Attune NxT flow cytometer (Invitrogen). Data was analyzed using the FlowJo software. Gates were created based on unstained control or isotype control for each sample.

## Immunofluorescence microscopy

GBM cells were analyzed for vaccinia infection and tumor antigen expression by immunofluorescence microscopy. GBM cells ($1 \times 10^5$) were plated in individual wells of 24-well glass-bottom plates (Cellvis) and incubated for 24 h to allow cells to adhere. Media were aspirated from each well, and cells were gently washed with phosphate-buffered saline (PBS; pH 7.4) + 1% bovine serum albumin. Cells were stained with primary CD19 antibody (Invitrogen) or EGFRvIII antibody (Novus Biologicals) at 4 °C for overnight, washed, and stained with secondary Alex 647-conjugated anti-mouse IgG (H + L) (Jackson ImmunoResearch) for 1 h at room temperature. Cells were then fixed and permeabilized using Cytofix/Cytoperm Fixation/Permeabilization Solution (BD Biosciences, Cat# 554714) for 10 min at room temperature. The solution was aspirated and stained with anti-vaccinia antibody (Abcam) for 1 h at room temperature, washed, and stained with Alexa 488-conjugated anti-rabbit IgG (H + L) (Jackson ImmunoResearch) for 1 h. Cells were counterstained with 4′6-diamidino-2-phenylindole (DAPI) for 5 min at room temperature, and each well was imaged at ×10 to ×63 magnification. Confocal microscopy was performed on a Zeiss LSM900 microscope (Zeiss). Alternatively, image was acquired using a Nikon ECLIPSE Ti2 inverted microscope (Nikon).

The detailed information of all the antibodies used was listed in Table S1.

## Luciferase reporter assay for cytotoxicity evaluation

For luciferase assay, GBM cells that expressed a luciferase reporter were dissociated with Accutase, seeded onto a 96-well plate with round-bottom (Greiner Bio-One, Cat#655090) one day before coculture. On the following day, GBM cells were infected with oncolytic virus for 4-6 h, and the BiCAR cells were added into the wells. After 12, 24, or 48 h of coculturing, luciferase activity in different wells were analyzed using the Luciferase Assay System (Promega, Cat# E6120).

## Enzyme-linked immunosorbent assay (ELISA) for IFNγ

To detect IFNγ levels in the culture medium by ELISA, tumor cells were seeded onto 96-well round-bottom plates (FALCON, Cat# 35922025) and infected with OVDual at the indicated MOI. After incubation at 37 °C for 4 h, the same number of BiCAR-T effector cells and tumor cells were added. 24 h post treatment by BiCAR-T, the conditioned medium was collected and analyzed using the human IFNγ ELISA kit (R&D, Cat# P310910). Plates were read at 450 nm using a Mini Max 300 Imaging Cytometer Microplate Reader (SPECTRA MAX, Cat# 309710).

## 3D tumor spheroid coculture assays

3D tumor spheroids were prepared by seeding Accutase-dissociated single GBM cells (PBT003-RFP) into ultra-low adhesion V bottom 96-well plates, cultured in DMEM/F12 medium (Gibco, Cat# 100-15), and added to 1% B27 (Thermo Fisher, 17504044), 10 pg/mL human recombinant fibroblast growth factor (FGF) (Thermo Fisher, Cat# 100-18B) and 10 pg/mL human recombinant epidermal growth factor (EGF) (Thermo Fisher, Cat# 100-15) for 48 h. For the T cell infiltration and chemotaxis experiments, 3D tumor spheroids of PBT003-RFP cells were infected with OVDual at an MOI of 1 for 5 h. BiCAR-T or control T cells ($5 \times 10^4$) were then added for coculture.

## Western blot

Western blots were performed as described[76]. Information on primary and secondary antibodies is listed in Table S1.

## Immunostaining of brain sections

Immunocytochemical analysis was carried out on PFA-fixed tissues following established protocols[49]. Briefly, brain tissues were collected from mice after perfusion. The harvested brains were post-fixed in 4% PFA and subsequently cryoprotected in 30% sucrose. Once cryoprotected, the brains were flash-frozen and stored at -20 °C. Serial sagittal cryosections were prepared for further analysis. For immunostaining, antigen retrieval was performed by treating the brain sections with citrate buffer (Sigma Aldrich, Cat# C9999), which was heated to boiling and then allowed to cool at room temperature (RT) for 1 h. Sections were permeabilized with PBST (PBS with 0.1% Triton-X100), followed by blocking in 5% donkey serum prepared in PBST for 1 h at RT. The sections were incubated for overnight at 4 °C with primary antibodies, anti-vaccinia (1:200), anti-hCD45 (1:800), anti-CD68 (1:200), or anti-IBA1 (1:200). After primary antibody incubation, the sections were washed in PBS and incubated with secondary antibodies, Cy5-conjugated anti-mouse IgG, Alexa 488-conjugated anti-rabbit IgG (1:100 each), 647-conjugated anti-goat IgG (1:100 each), or Alexa 488-conjugated anti-rat IgG (1:100 each) for 1 h at RT. Following secondary antibody staining, the sections were washed with PBS, counterstained with DAPI (1:10,000) for 10 minutes, and mounted using Fluoromount-G reagent (Southern Biotech, Cat# 0100-01). Information on primary and secondary antibodies is listed in Table S1. Imaging was performed using a Nikon Ti-2 inverted microscope. For each mouse, data from three brain slices were averaged to generate a single biological replicate. Data from $n = 3$ mice per group were subjected to statistical analysis.

## Transswell assay

Migration was assessed in 24-well plates with 5 mm pore size filters (Corning) and $5 \times 10^4$ tumor cells were infected with OV at an MOI of 1 for 5 h before BiCAR-T or control T cells ($1 \times 10^6$) were added to the upper chamber. After 24 h of incubation, cells migrated to the lower chamber were counted by flow cytometry using anti-CD3 antibody.

## In vivo tumor studies

For GBM cell transplantation, $2 \times 10^5$ GBM cells were transplanted into the frontal lobes of mouse brains (anterior posterior (AP) = + 0.6 mm, medial lateral (ML) = +1.6 mm, and dorsal ventral ( DV) = − 2.6 mm) by stereotaxic intracranial injection. For combination therapy, 1 week after GBM cell transplantation, tumor size was evaluated by bioluminescence imaging. Mice were treated with $6 \times 10^6$ PFU OV alone, $4 \times 10^6$ BiCAR-T or BiCAR-NK cells alone, the combination of the two agents, or no treatment. Tumor growth was monitored by bioluminescence imaging every week for 6 weeks. The bioluminescence intensity was quantified using Spectral Instruments Imaging-AMIView 1.7.061. Mice were euthanized when one or more of the early euthanasia criteria (failure to eat food or drink water for 24 h; failure to make normal postural adjustments or display normal behavior; obvious distress, such as hunched posture or unresponsiveness, or weight loss > 20%) were met. The survival of mice was recorded.

## RNA-seq data analysis

Total RNA was extracted from BiCAR-T cells 48 h after treatment using TRIzol. RNA deep sequencing analysis was carried out as previously published[85]. Briefly, NEBNext Ultra II (Directional) with Poly A selection were used to construct polyA-RNA-seq libraries for $2 \times 150$, 40 M paired end reads (20 M in each direction) runs per sample. Sequencing was performed on Illumina NovaSeq X Plus instrument by Admera health (https://www.admerahealth.com/). Fastq files obtained from Admera were mapped to the human hg38 genome using the subread aligner and read counting of genes was obtained with featureCounts. Subread aligner and featureCounts in Subread package (release 2.0.6) were used with default parameters[86]. Raw read count summarized by featureCounts were use in DEseq2 (version 1.40.1) to obtain normalized counts and to generate the differentially expressed gene (DEG) results at the default setting [the Wald-test was applied to assess the p value for differential gene expression and the adjusted $p$ value (p-adj) was obtained by the Benjamini and Hochberg method][87]. R (version 4.5), R studio, and Bioconductor software and associated packages, STRING, and Metascape were used for data analysis and visualization.

## Statistical analysis

Data are presented as means ± SD. Statistical comparisons were performed using the unpaired two-tailed Student's t test, one-way ANOVA test, or two-way ANOVA test, unless otherwise stated. Statistical comparison of Kaplan-Meier survival data was performed using the log-rank (Mantel-Cox) test.

## Reporting summary

Further information on research design is available in the Nature Portfolio Reporting Summary linked to this article.

## Data availability

The RNA-seq dataset has been uploaded to GEO database with GEO# GSE310003 (https://www.ncbi.nlm.nih.gov/geo/query/acc.cgi). Source data for all figures and Supplementary Figs. are provided within the paper.

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

## Acknowledgements

The authors would like to thank Louise and Herbert Horvitz for their generosity and forethought, Dr. Dario Campana from National University of Singapore for sharing the mbIL15 vector, and Dr. Carl June from University of Pennsylvania for sharing the EGFRvIII CAR vector. Research reported in this publication was also supported by the National Cancer Institute of the National Institutes of Health under award number P30CA33572. The content is solely the responsibility of the authors and does not necessarily represent the official views of the National Institutes of Health. BioRender has been used for preparing schematics.

## Author contributions

Conceptualization, Y.S. and J.L.; methodology, Y.S., J.L., S.C.; data acquisition and analysis, J.L., G.S.; investigation, J.L.; experimental support, G.S., S.C., Q.C., T.Z., Y.Q., and P.Y.; resources, X.W., Y.F., and M.V.M.; manuscript, Y.S. and J.L.

## Competing interests

S.C. is a consultant to Imugene Ltd. Y.F. is a paid scientific consultant for Medtronics, LivsMed, Imugene, Boston Scientific, Vergent, Eureka Therapeutics; receives royalties for inventions from Merck, XDemics, and Imugene Ltd; and owns the patent for CF33-Ovs licensed to Imugene Ltd. M.V.M. is an inventor on patents related to adoptive cell therapies, held by Massachusetts General Hospital (some licensed to Promab and Luminary) and University of Pennsylvania (some licensed to Novartis). M.V.M. receives Grant/Research support from BMS, Kite Pharma, Sobi. M.V.M. holds Equity in Altido Therapeutics. M.V.M. is a compensated Consultant for A2Bio, Adaptimmune, Alexion, Astellas, AstraZeneca, BMS, Cabaletta Bio, In8bio, KSQ, and Lumicks.
