## [Transparent Peer Review file · Nature Communications]

Developing a multimodal therapy for glioblastoma using bi-specific CARs and oncolytic virus delivering CD19 and EGFRvIII antigens

Corresponding Author: Professor Yanhong Shi

Version 0:

Reviewer comments:

Reviewer #1

(Remarks to the Author)

Please see PDF for version with figures. Below is text only:

This is overall an ambitious, complex multimodal approach to treating a difficult tumor, where a complex multimodal approach may be needed. Due to the complexity of the therapy, unfortunately more controls are needed and more assays are needed to really verify the therapy is working in the manner it is intended – especially in vivo. My major concerns are related to inadequately verifying many aspects of the basic mechanisms of action of the therapy, including:

Major Concerns

1. BiCAR Engineering and Functionality

Engineering bispecific CARs is inherently challenging, as the new construct must be properly trafficked to the cell surface, maintain correct folding, and allow for independent recognition and signaling after recognition by each antigen-binding domain. The paper does not rigorously demonstrate that the BiCAR created is successfully expressed on the surface of T or NK cells, nor that it can trigger effective tumor cell killing in response to either CD19 or EGFRvIII alone. The study does not directly demonstrate that the BiCAR is present on the surface of T or NK cells. Instead, it relies on CD34 as a marker, which does not confirm that the CAR itself is expressed or functional. Direct staining for the CAR, using an antibody specific to the CAR's extracellular domain (anti-linker antibody is a good one), is necessary to confirm proper surface localization. Without this, it is unclear whether the engineered cells are even capable of engaging with tumor antigens.

2. Inadequate Cytotoxicity (CTL) Assays

The cytotoxicity assays presented do not convincingly demonstrate CAR-specific killing. A rigorous CTL assay should include a range of effector-to-target (E:T) ratios such that at low E:T ratios, killing approaches the level of tumor-only controls, which would indicate specificity and potency that changes with E:T ratios. In the current data, all CTL curves—including experimental and control groups—have similar slopes, suggesting that the observed effects may be due to technical artifacts, such as uneven plating or pipetting, rather than true CAR-mediated cytotoxicity. The lack of a clear dose-response and the absence of data using single-antigen-positive targets further weakens the claim that the BiCAR is responsible for the observed effects.

3. Oncolytic Virus (OV) Antigen Delivery In Vivo

The effectiveness of the oncolytic virus in delivering both CD19 and EGFRvIII antigens to tumor cells in vivo is not demonstrated. There is no immunohistochemistry (IHC) or other direct evidence showing that these antigens are robustly and uniformly (or even partially) expressed on tumor cells after OV treatment. IHC or IF for Vaccinia is insufficient. This is critical, as the therapeutic strategy depends on reliable antigen delivery to sensitize tumor cells to BiCAR-T or BiCAR-NK cell targeting of those antigens. Without in vivo confirmation of successful delivery and antigen expression, the platform's efficacy cannot be properly evaluated.

4. Critical Unanswered Questions

Several key questions remain unresolved, undermining the study's conclusions:

o Is the dual CAR construct capable of mediating effective killing of tumor cells in vivo based on CD19 or EGFRvIII expression alone?

o How robustly and uniformly are CD19 and EGFRvIII antigens expressed on tumor cells in vivo after OV delivery of these antigens?

o Is the BiCAR construct stably and functionally expressed on the surface of T and NK cells, and does it trigger signaling and cytotoxicity in response to each antigen independently? Can the BiCAR kill tumor cells that are only CD19-positive or only EGFRvIII-positive?

5. Role of Membrane-Bound IL-15 and IL-21 and Lack of Proper In Vivo Controls

Membrane-bound IL-15 and IL-21 are known to be potent stimulators of T-cell or NK cell proliferation, survival, and cytotoxicity, even in the absence of a CAR on the T-cell or NK cell. The in vivo studies do not include appropriate controls—such as irrelevant-CAR T-cells or NK cells exposed to the same cytokine environment—to determine whether the observed anti-tumor effects are due to the CAR or simply the result of cytokine-driven T-cell/immune cell activation. Without these controls, it is impossible to attribute any therapeutic benefit specifically to the CAR, as opposed to the general immune stimulation provided by IL-15 and IL-21.

6. As described below, in vivo studies are low powered, with no significant differences observed in the objective measure of BLI, but slight significance shown with overall survival (which incorporates a subjective measure of when to sacrifice an animal). BLI is shown for a very short timeframe, making it difficult to assess the meaningfulness of the survival differences.

In summary, the study lacks critical data to demonstrate that the BiCAR construct is properly expressed and functional, both in vitro and in vivo. The cytotoxicity assays are not sufficiently rigorous, there is no direct evidence for effective antigen delivery in vivo or CAR surface expression on T or NK cells, and the contribution of the CAR itself is not distinguished from the effects of cytokine stimulation. These gaps must be addressed to validate the platform's therapeutic potential.

Additional specific comments, some related to the above general summary, are listed below:

Figure 3C: BLI is only shown for 6 weeks but survival is shown for 17 weeks. Please show the additional BLI weeks. Also, it looks like the survival curve is truncated right before the remaining 3 mice died. Please include the survival curve until death of the last 3 mice. In NSG mice, a more appropriate control for the CAR T cell group than OVDual virus alone is OVDual + control (non-targeting CAR, or truncated CAR) transduced T-cells.

“Mice treated with OVDual and BiCAR-T also reduced tumor progression compared to mice treated with OVDual alone, with a more dramatic reduction at weeks 5 and 6, although the difference did not reach statistical significance” – since there is no significant difference between the groups, it is misleading to say it reduced tumor progression – this should be removed rather than overstated.

In these CTLs (and all others throughout), as mentioned above, the E:T ratio should go down to a point where the killing is the same as it is with no effector present. Otherwise the specificity of the CAR cannot be assessed.

Fig 6 C and D. Same comment as for figure 3c. Please show longer luciferase time course, since the mice were followed for longer. 6D also appears to have been cut short just as OVDual+OVmL15/21+BiCAR started to die – please report a little longer until they die.

This is also missing OVDual+OVmL15/21+control CAR T cells, which is required to show that there is a CAR-specific effect. Same comments for Figure 8 as well, which has the same major limitations.

Fig 7B, 7D, and in T-cell figures: “BiCAR” does not indicate surface CAR expression.

Staining for the CD34 tag on the BiCAR vector is very different from staining for the actual BiCAR, which can be done in numerous ways (anti-linker antibody is generally easiest).

Fig 2E is missing the OVDual bar. Also a 48 hour time point would be helpful.

Reviewer #2

(Remarks to the Author)

This study by Shi et al. describes a compelling multimodal strategy integrating oncolytic virotherapy (OVDual and OVmL15/21) with BiCAR-T/NK cells to overcome key limitations in GBM treatment. The approach effectively enhances tumor antigen presentation, immune activation, and anti-tumor efficacy in both in vitro and in vivo GBM models. The strategy allows to extend the therapeutic opportunities to a larger population of patients affected by GBM. Overall, this is relevant in the field of neuro-oncology, but there are some major concerns that should be addressed.

1. The benefit of BiCAR-T/NK remains unsubstantiated. In the manuscript, the authors did not demonstrate the therapeutic relevance of delivering two target antigens: hEGFRvIII and hCD19t. Is that necessary? Would be more convincing if the authors compared the overall survival of the OVDual/BiCAR treatment vs the OV-mono antigen/mono CAR treatment (e.g. OV-hEGFRvIII / EGFRvIII scFv CAR)

2. In the GBM mouse model, as additional control can be included BiCAR-T treatment alone to rule out “off-target” effects, as demonstrated in vitro (Fig. 2). I would suggest N to be consistent (in Fig. 3, N=4, while in Fig. 6/8, N=5). Furthermore, a larger sample size would increase the statistical robustness of the study.

3. In Fig. 5 and Fig. S3, the study on the modulation of the tumor microenvironment is somewhat limited. The authors should

provide a more comprehensive analysis of the TME and the activation state of macrophages and microglia among the different conditions.

4. In Fig. 6 and Fig. 8, treatments with OVDual+OVmIL15/21+BiCAR-T/NK extend the overall survival, however the authors do not provide any insights on the activation of key downstream pathways dependent on IL15/21 signalling. Also, they do not provide key phenotypic differences between the treatments OVDual+OVmIL15/21+BiCAR-T/NK and OVDual+OVGFP+BiCAR-T/NK.

Overall interesting study that could enhance the targetability of GBM tumours. However, not developed enough to grant publication as critical controls and key mechanistic insights are missing to substantiate the validity of the approach.

Reviewer #3

(Remarks to the Author)

This study presents an innovative approach to treating glioblastoma by combining bispecific CAR-T (BiCAR-T) cells with oncolytic viruses encoding dual antigens (CD19t and EGFRvIII) and membrane-bound cytokines (mIL15/21). The platform is rigorously tested in vitro and in vivo, showing potential to overcome antigen heterogeneity and the immunosuppressive tumor microenvironment. While the manuscript is well-structured and addresses an important clinical problem, several specific experimental additions are needed to support the claims and strengthen the conclusions.

Figure 1: The design of OVDual and BiCAR-T is logical and well-illustrated. However, additional validation could demonstrate whether the expression of CD19t and EGFRvIII on GBM cells translates into functional CAR engagement.

- Include downstream signaling readouts in BiCAR-T (e.g., phospho-ZAP70, CD69, or cytokine production upon antigen stimulation).

Figure 2: The cytotoxicity results are compelling, but OV-mediated killing occurs at 48h.

- The OVDual-alone group (no T cells) shows limited killing at early time points but, by 48 hours, reaches a level of cytotoxicity comparable to the BiCAR-T + OVDual group. How do you explain this?

- Show statistics in Fig 2C.

- Fig 2E: color of bars do not match the legends'.

Figure 3: While the survival benefits are convincing, tumor volume reduction (Fig. 3C) is unclear in terms of statistical significance. Suggested additions:

- Reframe the text to reflect statistical outcomes.

- Provide fold change or % reduction in luciferase signal at key time points for each group to clarify the magnitude of tumor control.

- Include a BiCAR-T alone group to isolate each component's in vivo effect.

Figure 4h:

- Instead of saying "marked increase," give the numerical fold change (e.g., "a 1.8-fold boost in killing with OVmIL15/21 vs. OVGFP").

- Though you include BiCAR-T alone, a condition with uninfected GBM + BiCAR-T + OVmIL15/21 (no antigen delivery) would confirm that the cytokine effect alone doesn't drive non-specific killing.

- Report both statistical significance and actual % differences in killing (e.g., 35% vs. 25%) to contextualize the magnitude of enhancement.

- Use brackets or legend notes to clarify that asterisks indicate comparisons between OVmIL15/21 and OVGFP groups.

Figures 5 & S3: These figures show OV spread, T cell presence, and microglial activation via immunofluorescence. However, all data are qualitative.

- Quantify % area or mean intensity of vaccinia, hCD45, IBA1, and CD68 signals per brain section.

- Include statistical comparisons across treatment groups.

Figure 7: The workflow is clear, but BiCAR signaling in NK cells is not functionally validated.

- After antigen stimulation, measure CD107a, Granzyme B, or IFN- γ production to confirm BiCAR-NK functional activation.

Figure 8: While the combination shows robust tumor control, several mechanistic elements are missing.

- Track BiCAR-NK persistence and tumor infiltration (e.g., via labeled cells or hCD45/CD56 staining in brain sections).

- Quantify tumor burden reduction as fold change or % relative to control.

Reviewer #4

(Remarks to the Author)

Version 1:

Reviewer comments:

Reviewer #1

(Remarks to the Author)

This is overall a much improved version of the manuscript, with my previous comments addressed, and only a small number

of points remaining:

Major Point relating to multiple figures (Fig. 5, S7, S13): Pseudoreplication in Statistical Analysis

It appears that the statistical comparisons have been performed using individual brain slices as independent data points (n=9 per group, derived from 3 slices per mouse across an n of 3 mice). This approach constitutes pseudoreplication and artificially inflates your sample size, leading to an underestimation of variance and potentially spurious statistical significance (low p-values).

Your true biological unit of replication is the individual mouse, not the brain slice. Slices taken from the same animal are technical replicates; they are not independent observations as they share the same genetic background, environmental factors, and treatment history. Treating them as independent violates the assumptions of most standard statistical tests and can lead to erroneous conclusions.

To address this critical issue, I strongly recommend the following:

1. Aggregate Data at the Biological Replicate Level: For each mouse, calculate a single average value (e.g., percent) for each cellular quantification across its three brain slices. This will provide one biological data point per mouse.
2. Perform Statistics on Mouse-Level Data: Conduct all primary statistical analyses using these aggregated mouse-level values. This means your sample size for each experimental group will be $n = 3$ (representing the three individual mice), not $n = 9$. It looks like this is likely still significant for many of the figures, but regardless of significance, it is more accurate.
3. Adjust Presentation of Data:

o In your figures, if you wish to show the variability within mice (i.e., the individual slice data), please do so in a manner that clearly indicates the nesting (e.g., using distinct symbols or colors for each mouse, or connecting lines between slices from the same mouse). However this does not change the statistical tests. Ensure that the statistical tests and reported n values in your methods and results sections accurately reflect the number of biological replicates (mice; ie 3).

Minor point: The statement "Although the survival benefit of BiCAR-T cells combined with OVDual was modest in the xenograft NSG model, the implications of this finding are likely far more impactful in the clinical setting" is an overstatement as often these therapies are less impactful in the clinical setting; revise wording to not speculate beyond what the data show.

Reviewer #2

(Remarks to the Author)

The authors have addressed, although not in full, my previous concerns. I appreciate the effort made to strengthen the manuscript by incorporating the requested in vitro and in vivo controls. These additions certainly improve the technical rigor of the study.

That said, the critical issue remains the very modest therapeutic efficacy observed with the BiCAR approach compared to the mono-CAR treatment (Figure S6). While the multimodal strategy is conceptually interesting and may increase targetability in the context of antigen escape in GBM, the survival benefit demonstrated here is limited. It remains unclear whether this level of improvement justifies the added complexity of the proposed multimodal platform.

Moreover, the authors' comment "the implications of this finding are likely more impactful in the clinical setting, because NSG mice are immunodeficient and do not recapitulate the complex human GBM tumor microenvironment, which includes diverse immune population, cytokine network, and selective pressure that drive dynamic changes in antigen expression." remains highly speculative.

The translational path for combining dual-antigen oncolytic viruses with BiCAR-T/NK cells appears challenging. Manufacturing complexity, regulatory burden, and clinical implementation may significantly limit its applicability, particularly in light of the modest efficacy observed. Overall, while the strategy is innovative, stronger evidence of therapeutic advantage would be needed to support its relevance and competitiveness in the GBM cell therapy landscape.

Reviewer #3

(Remarks to the Author)

The authors have answered all my questions

Reviewer #4

(Remarks to the Author)

Response to reviewer comments (NCOMMS-25-23521-T)

Reviewer #1

This is overall an ambitious, complex multimodal approach to treating a difficult tumor, where a complex multimodal approach may be needed. Due to the complexity of the therapy, unfortunately more controls are needed and more assays are needed to really verify the therapy is working in the manner it is intended – especially in vivo. My major concerns are related to inadequately verifying many aspects of the basic mechanisms of action of the therapy, including:

Response: We thank the reviewer for acknowledging that “this is overall an ambitious, complex multimodal approach to treating a difficult tumor, where a complex multimodal approach may be needed.” We have now provided additional controls and assays as suggested. The detailed response is included below.

Major Concerns

1. BiCAR Engineering and Functionality

Engineering bispecific CARs is inherently challenging, as the new construct must be properly trafficked to the cell surface, maintain correct folding, and allow for independent recognition and signaling after recognition by each antigen-binding domain. The paper does not rigorously demonstrate that the BiCAR created is successfully expressed on the surface of T or NK cells, nor that it can trigger effective tumor cell killing in response to either CD19 or EGFRvIII alone. The study does not directly demonstrate that the BiCAR is present on the surface of T or NK cells. Instead, it relies on CD34 as a marker, which does not confirm that the CAR itself is expressed or functional. Direct staining for the CAR, using an antibody specific to the CAR’s extracellular domain (anti-linker antibody is a good one), is necessary to confirm proper surface localization. Without this, it is unclear whether the engineered cells are even capable of engaging with tumor antigens.

Response: We have now shown direct staining for CAR using the anti-linker antibody as suggested. The result showed 99.7% of anti-linker-positive cells on BiCAR-T cells (Fig. 1G lower panel), confirming that BiCAR is present on BiCAR-T cells. Similar results were obtained for BiCAR-NK cells (Fig. 7D), confirming that BiCAR is present on BiCAR-NK cells. Moreover, we have now shown that BiCAR-T exhibited effective tumor killing in response to either CD19 or EGFRvIII antigen alone (Fig. S2 and Fig. S5). Thank you for your great suggestion of including these important controls, which has strengthened our study substantially.

2. Inadequate Cytotoxicity (CTL) Assays

The cytotoxicity assays presented do not convincingly demonstrate CAR-specific killing. A rigorous CTL assay should include a range of effector-to-target (E:T) ratios such that at low E:T ratios, killing approaches the level of tumor-only controls, which would indicate specificity and potency that changes with E:T ratios. In the current data, all CTL curves—including experimental and control groups—have similar slopes, suggesting that the observed effects may be due to technical artifacts, such as uneven plating or pipetting, rather than true CAR-mediated cytotoxicity. The lack of a clear dose-response and the absence of data using single-antigen-positive targets further weakens the claim that the BiCAR is responsible for the observed effects.

Response: We have now performed a range of E:T ratios including 0.25 to 1, 0.5 to 1, and 1 to 1 as suggested. At the lower E:T ratio (0.25 to 1), the killing effect of OVDual+BiCAR-T approached the level of OVDual alone or OVDul+Control T cells (Fig. S3). Moreover, we were able to see a

clear dose response for the treatment of OVDual+BiCAR-T with the E:T ratio of 0.25:1, 0.5:1, and 1:1 (Fig. S3). In addition to OVDual+BiCAR-T condition, we were also able to see dose-dependent killing when BiCAR-T was combined with single antigen EGFRvIII or CD19 (Fig. S2A). Additional killing data for the combination of BiCAR-T with single antigens CD19 or EGFRvIII were included in Fig. S2B.

3. Oncolytic Virus (OV) Antigen Delivery In Vivo

The effectiveness of the oncolytic virus in delivering both CD19 and EGFRvIII antigens to tumor cells in vivo is not demonstrated. There is no immunohistochemistry (IHC) or other direct evidence showing that these antigens are robustly and uniformly (or even partially) expressed on tumor cells after OV treatment. IHC or IF for Vaccinia is insufficient. This is critical, as the therapeutic strategy depends on reliable antigen delivery to sensitize tumor cells to BiCAR-T or BiCAR-NK cell targeting of those antigens. Without in vivo confirmation of successful delivery and antigen expression, the platform's efficacy cannot be properly evaluated.

Response: We have now performed immunohistochemistry to show that the hEGFRvIII and hCD19 antigens are expressed on tumor cells after OVDual treatment, and treatment with BiCAR-T together with OVDual led to substantially increased spreading of OV-delivered antigens (Fig. S7A-D).

4. Critical Unanswered Questions

Several key questions remain unresolved, undermining the study's conclusions:

- o Is the dual CAR construct capable of mediating effective killing of tumor cells in vivo based on CD19 or EGFRvIII expression alone?

Response: Yes, the BiCAR-construct was able to mediate effective killing of tumor cells infected with oncolytic virus with single hEGFRvIII or hCD19 antigen alone in vivo. Tumor progression was reduced substantially in mice treated with OVEGFRvIII+BiCAR-T or OVCD19+BiCAR-T, compared to mice treated with OVGFP+BiCAR-T or received no treatment, as shown in Fig. S5. Accordingly, the survival of tumor-bearing mice was prolonged in mice treated with OVEGFRvIII+BiCAR-T or OVCD19+BiCAR-T, compared to mice treated with OVGFP+BiCAR-T or received no treatment, as shown in Fig. S5.

- o How robustly and uniformly are CD19 and EGFRvIII antigens expressed on tumor cells in vivo after OV delivery of these antigens?

Response: We have now performed immunohistochemistry to show that the EGFRvIII and CD19 antigens are expressed on tumor cells after OVDual treatment, and treatment with BiCAR-T together with OVDual led to substantially increased spreading of OV-delivered antigens, reaching about 80% tumor cells 14 days after OVDual and BiCAR-T treatment (Fig. S7A-D).

- o Is the BiCAR construct stably and functionally expressed on the surface of T and NK cells, and does it trigger signaling and cytotoxicity in response to each antigen independently? Can the BiCAR kill tumor cells that are only CD19-positive or only EGFRvIII-positive?

Response: Yes, the BiCAR constructs were expressed on the surface of BiCAR-T (Fig. 1G) and BiCAR-NK cells (Fig. 7D) as revealed by flow cytometry analysis using an antibody for the G4S linker as the reviewer suggested in comments 1.

And yes, BiCAR-T triggered signaling and cytotoxicity in response to the hCD19 or hEGFRvIII antigen alone (Fig. S2A-C). Moreover, BiCAR-T killed tumor cells that are infected with oncolytic

virus encoding hEGFRvIII^t or hCD19^t antigen alone (OVEGFRvIII^t or OVCD19^t) in vivo (Fig. S5). See also response to comment 4 above.

5. Role of Membrane-Bound IL-15 and IL-21 and Lack of Proper In Vivo Controls
Membrane-bound IL-15 and IL-21 are known to be potent stimulators of T-cell or NK cell proliferation, survival, and cytotoxicity, even in the absence of a CAR on the T-cell or NK cell. The in vivo studies do not include appropriate controls—such as irrelevant-CAR T-cells or NK cells exposed to the same cytokine environment—to determine whether the observed anti-tumor effects are due to the CAR or simply the result of cytokine-driven T-cell/immune cell activation. Without these controls, it is impossible to attribute any therapeutic benefit specifically to the CAR, as opposed to the general immune stimulation provided by IL-15 and IL-21.

Response: We have now included control T cells exposed to the same cytokines as a negative control as suggested. We showed that when mice were treated with control T cells plus OVDual and OVmIL15/21 vs control T cells plus OVDual and OVGFP, there is no statistically significant difference in tumor progression (Fig. S11). In contrast, there is dramatically reduced tumor progression in mice treated with BiCAR-T cells plus OVDual and OVmIL15/21, compared to mice treated with BiCAR-T cells plus OVDual and OVGFP (Fig. 6B and C). Moreover, we detected reduced tumor progression in mice treated with BiCAR-T plus OVDual and OVmIL15/21, compared to mice treated with control T plus OVDual and OVmIL15/21 (Fig. S11B and C). These results together indicate that the OVmIL15/21-boosted anti-tumor effect is dependent on BiCAR on T cells.

6. As described below, in vivo studies are low powered, with no significant differences observed in the objective measure of BLI, but slight significance shown with overall survival (which incorporates a subjective measure of when to sacrifice an animal). BLI is shown for a very short timeframe, making it difficult to assess the meaningfulness of the survival differences.

Response: Thank you for the comment. We believe the reviewer referred to the small n and lack of statistically significant difference in tumor progression in original Fig. 3. We have now repeated the experiment with increased mouse number (n=5). We are able to detect statistically significant decrease of tumor progression and improvement in animal survival in mice treated with OVDual plus BiCAR-T vs mice treated with OVDual alone or BiCAR-T alone or no treatment (Fig. 3).

We have extended all relevant bioluminescence imaging (BLI) analysis to 6 weeks because most untreated mice began to die around week 6. Moreover, the tumor signals in untreated animals beyond week 6 became saturated under the same exposure settings used for treated groups. Therefore, we did not include BLI images beyond week 6. We have updated Figure 6 and Figure 8, repeated Figure 3, and generated new supplementary figures (Figure S5, Figure S6, and Figure S11) to present the full 6-week BLI data.

In summary, the study lacks critical data to demonstrate that the BiCAR construct is properly expressed and functional, both in vitro and in vivo. The cytotoxicity assays are not sufficiently rigorous, there is no direct evidence for effective antigen delivery in vivo or CAR surface expression on T or NK cells, and the contribution of the CAR itself is not distinguished from the effects of cytokine stimulation. These gaps must be addressed to validate the platform's therapeutic potential.

Response: We thank the reviewer for critical evaluation of our manuscript. We have now provided additional data to address each of these points as detailed above.

Additional specific comments, some related to the above general summary, are listed below: Figure 3C: BLI is only shown for 6 weeks but survival is shown for 17 weeks. Please show the additional BLI weeks. Also, it looks like the survival curve is truncated right before the remaining 3 mice died. Please include the survival curve until death of the last 3 mice. In NSG mice, a more appropriate control for the CAR T cell group than OVDual virus alone is OVDual + control (non-targeting CAR, or truncated CAR) transduced T-cells.

Response: For Fig. 3C: We included BLI for 6 weeks because most mice with no treatment died before or around the end of 6 weeks. Moreover, by 6 weeks, the BLI signal for mice without treatment becomes saturated under the same exposure settings used for treated groups. We could not detect much increase in BLI signals over time in the no-treatment control group beyond 6 weeks. Therefore, the comparison of other groups with the no-treatment control group could not be done properly after 6 weeks.

We have repeated the experiment and included the survival curve until the death of all mice now (Fig. 3D). Because we have compared OVDual and OVDual + control T in killing tumor cells in vitro and shown that OVDual and OVDual + control T cells exhibited comparable tumor killing effect (Fig. 2B and Fig. S3), we did not include OVDual + control T cells in Fig. 3.

“Mice treated with OVDual and BiCAR-T also reduced tumor progression compared to mice treated with OVDual alone, with a more dramatic reduction at weeks 5 and 6, although the difference did not reach statistical significance” – since there is no significant difference between the groups, it is misleading to say it reduced tumor progression – this should be removed rather than overstated.

Response: We agree with the reviewer and have repeated the experiment by including more mice per group. The result now showed statistically significant difference in tumor progression between OVDual alone and OVDual+BiCAR-T groups (Fig. 3B and C).

In these CTLs (and all others throughout), as mentioned above, the E:T ratio should go down to a point where the killing is the same as it is with no effector present. Otherwise the specificity of the CAR cannot be assessed.

Response: We have now performed CTL assays using different E:T ratios as suggested and shown a dose-dependent killing effect by BiCAR-T cells, and at the lower E:T ratio of 0.25:1, the killing effect by OVDual and BiCAR-T is close to that by OVDual alone or OVDual and control T cells (Fig. S3).

Fig 6 C and D. Same comment as for figure 3c. Please show longer luciferase time course, since the mice were followed for longer. 6D also appears to have been cut short just as OVDual+OVmIL15/21+BiCAR started to die – please report a little longer until they die. This is also missing OVDual+OVmIL15/21+control CAR T cells, which is required to show that there is a CAR-specific effect. Same comments for Figure 8 as well, which has the same major limitations.

Response: For Fig. 6C, we have increased BLI from 4 weeks to 6 weeks. By 6 weeks, the BLI signal for mice without treatment becomes saturated under the same exposure settings used for treated groups. We could not detect much increase in signals over time in the no-treatment control group beyond 6 weeks, thus the comparison of other groups with the no-treatment group could not be done properly after 6 weeks. Therefore, we decided to show BLI images up to week 6.

For Fig. 6D, we have included the survival curve until all mice died in revised Fig. 6D (all remaining mice from the OVDual+OVmIL15/21+BiCAR-T group were euthanized around day 120).

Moreover, we have now included OVDual+OVGFP+control T cell group and OVDual+OVmIL15/21+control T cell groups in Fig. S11 now. Mice treated with OVDual+OVGFP+control T cells vs OVDual+OVmIL15/21+control T cells did not exhibit statistically significant difference in tumor progression (Fig. S11B and C). In contrast, mice treated with OVDual+OVmIL15/21+BiCAR-T exhibited substantially reduced tumor progression than mice treated with OVDual+OVmIL15/21+control T (Fig. S11B and C) or OVDual+OVGFP+BiCAR-T (Fig. 6B-D), indicating that the cytokine-boosted anti-tumor effect is CAR-dependent.

For Fig. 8C, we have increased BLI from 4 weeks to 6 weeks. By 6 weeks, the BLI signal for mice without treatment becomes saturated under the same exposure settings used for treated groups. We could not detect further increase in the signals over time anymore in the no treatment control group. Therefore, the comparison of other groups with the no treatment control group could not be done properly after 6 weeks. We have now included the survival curve until all mice died in Fig. 8D.

Fig 7B, 7D, and in T-cell figures: “BiCAR” does not indicate surface CAR expression. Staining for the CD34 tag on the BiCAR vector is very different from staining for the actual BiCAR, which can be done in numerous ways (anti-linker antibody is generally easiest). **Response:** We have now shown direct staining for CAR in BiCAR-hESC cells (Fig. 7B, lower panel) or in BiCAR-NK cells (Fig. 7D, right panel) using the anti-linker antibody as the reviewer suggested in major comment 1, confirming that BiCAR is present on BiCAR-hESC cells and BiCAR-NK cells.

Fig 2E is missing the OVDual bar. Also a 48 hour time point would be helpful.

Response: We mislabeled the OVDual bar and have corrected it now in revised Fig. 2E. We have also included the 48 hr data now in Fig. 2E (right panel) as suggested.

We thank the reviewer for the great suggestions and believe that the additional data we have provided now have addressed most of the reviewer’s comments and strengthened our manuscript substantially.

Reviewer #2

This study by Shi et al. describes a compelling multimodal strategy integrating oncolytic virotherapy (OVDual and OVmIL15/21) with BiCAR-T/NK cells to overcome key limitations in GBM treatment. The approach effectively enhances tumor antigen presentation, immune activation, and anti-tumor efficacy in both in vitro and in vivo GBM models. The strategy allows to extend the therapeutic opportunities to a larger population of patients affected by GBM. Overall, this is relevant in the field of neuro-oncology, but there are some major concerns that should be addressed.

Response: We thank the reviewer for the overall positive comments and acknowledging that “The approach effectively enhances tumor antigen presentation, immune activation, and anti-tumor efficacy in both in vitro and in vivo GBM models. The strategy allows to extend the therapeutic opportunities to a larger population of patients affected by GBM. Overall, this is relevant in the field of neuro-oncology”. We have now addressed the reviewer comments one-by-one in the following, which has strengthened our study substantially. Thank you.

1. The benefit of BiCAR-T/NK remains unsubstantiated. In the manuscript, the authors did not demonstrate the therapeutic relevance of delivering two target antigens: hEGFRvIII and hCD19t. Is that necessary? Would be more convincing if the authors compared the overall survival of the OVDual/BiCAR treatment vs the OV-mono antigen/mono CAR treatment (e.g. OV-hEGFRvIII / EGFRvIII scFv CAR).

Response: We thank the reviewer for raising a great question. We believe the benefit of BiCAR-T/NK lies in the following: While CAR-T treatment of hematologic malignancies has achieved significant success in patients with refractory B cell malignancies, the outcome of this strategy in patients with solid tumors, especially GBM, has not always met with success. To address the major challenges faced in cancer immunotherapy for GBM, including heterogeneous tumor antigen expression, antigen loss or escape, and the immunosuppressive tumor microenvironment, in this study, we developed a multimodal approach to target GBM by using BiCAR-T/NK in combination with OVDual encoding two different antigens.

As suggested, we have compared the overall survival of mice received the OVDual/BiCAR-T treatment vs the OVEGFRvIII/EGFRvIII CAR-T cells (Fig. 8). We were able to detect modestly improved survival in mice treated with OVDual/BiCAR-T vs that in mice treated with OVEGFRvIII/EGFRvIII CAR-T cells with a p value is 0.08 (Fig. S6C). Although the survival benefit of BiCAR-T cells combined with OVDual was modest in the xenograft NSG model, the implications of this finding are likely more impactful in the clinical setting, because NSG mice are immunodeficient and do not recapitulate the complex human GBM tumor microenvironment, which includes diverse immune population, cytokine network, and selective pressure that drive dynamic changes in antigen expression. In GBM patients, antigen heterogeneity and antigen loss are major barriers that limit the effectiveness of CAR-T therapy using a single CAR^{7,11,14-17}. Our dual-antigen-encoding OV provides an additional layer of protection by introducing a second, independent antigen, thereby reducing the likelihood that tumor cells will fully evade CAR-T targeting. When OVDual is paired with BiCAR-T cells capable of responding to either antigen, this strategy has the potential to prevent antigen escape and achieve more robust and durable anti-tumor response in the more complex human GBM microenvironment.

2. In the GBM mouse model, as additional control can be included BiCAR-T treatment alone to rule out “off-target” effects, as demonstrated in vitro (Fig. 2). I would suggest N to be consistent (in Fig. 3, N=4, while in Fig. 6/8, N=5). Furthermore, a larger sample size would increase the statistical robustness of the study.

Response: We agree with the reviewer and have included BiCAR-T treatment alone as an additional treatment (Fig. 3). We also increased n (from 4 to 5) in Fig. 3. Indeed, a larger “n” has allowed us to detect statistically significant decrease in tumor progression in mice treated with OVDual+BiCAR-T compared to mice treated with OVDual alone or BiCAR-T alone.

3. In Fig. 5 and Fig. S3, the study on the modulation of the tumor microenvironment is somewhat limited. The authors should provide a more comprehensive analysis of the TME and the activation state of macrophages and microglia among the different conditions.

Response: We thank the reviewer for the great suggestion. We have now provided more quantitative analysis of the TME and evaluated the activation state of microglia by quantifying the microglia tumor infiltration by the percentage of CD68+IBA1+ area in RFP+ tumor area. We

detected substantially elevated IBA1⁺ microglia and CD68⁺IBA1⁺ activated microglia in the tumor area in mice treated with OVDual+BiCAR-T plus a higher dose of OVmIL15/21 than that in mice treated with OVDual+BiCAR-T plus a lower dose of OVmIL15/21 (Fig. 5E-G). We also assessed BiCAR-T cell infiltration into RFP⁺ tumor area by quantifying the percentage of CD45⁺ area within the RFP⁺ tumor area (Fig. S7E), which showed that substantially more BiCAR-T cells infiltrated the tumor area in the presence of OVDual (Fig. S7E and G). Moreover, we observed a greater vaccinia virus-positive area within the RFP⁺ tumor area under the BiCAR-T + OVDual condition (Fig. S7E and 7F), indicating that BiCAR-T cells may further benefit vaccinia virus release and replication.

4. In Fig. 6 and Fig. 8, treatments with OVDual+OVmIL15/21+BiCAR-T/NK extend the overall survival, however the authors do not provide any insights on the activation of key downstream pathways dependent on IL15/21 signalling. Also, they do not provide key phenotypic differences between the treatments OVDual+OVmIL15/21+BiCAR-T/NK and OVDual+OVGFP+BiCAR-T/NK.

Response: To address the reviewer's question regarding downstream IL-15/21 signaling between treatment groups, we performed bulk RNA-seq of BiCAR-T cells following coculture with GBM cells infected with OVDual in combination with either OVmIL15/21 or OVGFP. BiCAR-T cells were collected 48 h after treatment, a time point at which tumor cells were eliminated, enabling analysis of cytokine-driven transcriptional change in BiCAR-T cells. RNA-seq revealed robust transcriptional change induced by OVmIL15/21. A total of 97 genes were significantly differentially expressed ($|FC| > 1.5$ and adjusted $P < 0.05$), with 45 genes upregulated and 52 genes downregulated in BiCAR-T cells treated with OVmIL15/21 compared to BiCAR-T cells treated with OVGFP (Fig. S10A). Pathway enrichment analysis revealed that the upregulated genes were predominantly associated with cell killing, chemotaxis, leukocyte activation, T cell-mediated immunity, immune effector process, chemokine signaling, leukocyte-mediated cytotoxicity, and positive activation of T cell activation (Fig. S10B), all of which could lead to enhanced tumor cytotoxicity by BiCAR-T cells. Further analysis of upregulated genes for reactome pathways revealed a strong enrichment in cytokine signaling pathways, including IL-15, IL-2, IL-3, IL-5, and GM-CSF signaling (Fig. S10C). The activation of IL-15 and IL-21 signaling pathways was further demonstrated by the upregulation of their canonical signaling component JAK3 and the shared receptor subunit IL2RB as shown in the targeted heatmap (Fig. S10D). This cytokine-driven state was further evidenced by the elevated expression of cytotoxic effector molecules PRF1 (perforin 1), GNLY (granulysin), and GZMH (granzyme H)⁵¹⁻⁵³ in BiCAR-T cells treated with OVmIL15/21 vs OVGFP (Fig. S10D), establishing a functional link between pathway activation and enhanced BiCAR-T cell cytolytic capacity. Together, these results demonstrate that OVmIL15/21 delivers functional membrane-bound cytokines to boost BiCAR-T tumor cytotoxicity by activating pathways related to leukocyte chemotaxis, leukocyte activation, and immune effector process.

In terms of phenotypical difference, we have shown statistically significant decrease of tumor luciferase activity in mice treated with OVDual+OVmIL15/21+BiCAR-T/NK vs OVDual+OVGFP+BiCAR-T/NK and statistically significant improvement of tumor-bearing mouse survival in mice treated with OVDual+OVmIL15/21+BiCAR-T/NK vs OVDual+OVGFP+BiCAR-T/NK in Fig. 6 and Fig. 8.

Overall interesting study that could enhance the targetability of GBM tumours. However, not developed enough to grant publication as critical controls and key mechanistic insights are missing to substantiate the validity of the approach.

Response: We thank the reviewer again for acknowledging that this is an interesting study that could enhance the targetability of GBM tumors. We have now performed additional experiments and provided critical controls and key mechanistic insights as suggested to substantiate the validity of the approach as detailed in the above.

Reviewer #3

This study presents an innovative approach to treating glioblastoma by combining bispecific CAR-T (BiCAR-T) cells with oncolytic viruses encoding dual antigens (CD19t and EGFRvIII_t) and membrane-bound cytokines (mIL15/21). The platform is rigorously tested in vitro and in vivo, showing potential to overcome antigen heterogeneity and the immunosuppressive tumor microenvironment. While the manuscript is well-structured and addresses an important clinical problem, several specific experimental additions are needed to support the claims and strengthen the conclusions.

Response: We thank the reviewer for the very positive comments and acknowledging our “innovative approach”, “rigorous platform”, “well-structured manuscript”, and “addressing an important clinical problem”. We have now addressed your comments one-by-one in the following, which has strengthened our study substantially.

Figure 1: The design of OVDual and BiCAR-T is logical and well-illustrated. However, additional validation could demonstrate whether the expression of CD19_t and EGFRvIII_t on GBM cells translates into functional CAR engagement.

- Include downstream signaling readouts in BiCAR-T (e.g., phospho-ZAP70, CD69, or cytokine production upon antigen stimulation).

Response: We thank the reviewer for the great suggestion. We have now shown activation of downstream signaling, including CD25, CD69, and CD107a, in BiCAR-T cocultured with OVDual-infected GBM cells (Fig. 2C), or OVCD19_t- or OVEGFRvIII_t-infected GBM cells (Fig. S2C). In contrast, these signaling molecules were not activated when BiCAR-T cells were cocultured with OVGFP-infected GBM cells (without antigens) (Fig. 2C and Fig. S2C).

Figure 2: The cytotoxicity results are compelling, but OV-mediated killing occurs at 48h.

- The OVDual-alone group (no T cells) shows limited killing at early time points but, by 48 hours, reaches a level of cytotoxicity comparable to the BiCAR-T + OVDual group. How do you explain this?

Response: The OVDual-alone group exhibited limited killing at early time points but much higher cytotoxicity by 48 hr because OV can amplify in tumor cells. After killing infected tumor cells, OV can be released and up taken by neighboring tumor cells to kill more tumor cells.

- Show statistics in Fig 2C.

Response: Statistics for original Fig. 2C are shown in revised Fig. 2B now.

- Fig 2E: color of bars do not match the legends:

Response: We thank the reviewer for pointing out this issue, which was caused by a mistake by us. We have fixed it now in revised Fig. 2E.

Figure 3: While the survival benefits are convincing, tumor volume reduction (Fig. 3C) is unclear in terms of statistical significance. Suggested additions:

- Reframe the text to reflect statistical outcomes.
- Provide fold change or % reduction in luciferase signal at key time points for each group to clarify the magnitude of tumor control.
- Include a BiCAR-T alone group to isolate each component's in vivo effect.

Response: We thank the reviewer for the great suggestion. We have now included BiCAR-T alone group in Fig. 3 as suggested. Moreover, we have included more mice in the experiment now, which has allowed us to detect statistically significant decrease in tumor luciferase activity in mice treated with OVDual+BiCAR-T vs mice treated with OVDual alone or BiCAR-T alone (Fig. 3).

As suggested, we have now quantified percent reduction of tumor luciferase activity relative to the untreated control and provided % reduction in luciferase signal at key time points for each group to clarify the magnitude of tumor control. For example, at week 1, % reduction of tumor luciferase activity relative to no-treatment control was -0.58% for BiCAR-T, 78.47% for OVDual alone, and 94.5% for OVDual+BiCAR-T group; and at week 2, % reduction of tumor luciferase activity relative to no-treatment control was 12.5% for BiCAR-T, 79.64% for OVDual alone, and 95.96% for OVDual+BiCAR-T group.

Figure 4h:

- Instead of saying “marked increase,” give the numerical fold change (e.g., “a 1.8-fold boost in killing with OVmIL15/21 vs. OVGFP”).

Response: Yes, instead of saying “marked increase”, we have now provided examples of % reduction in tumor cell luciferase activity as we did for Fig. 3. For example, % reduction in tumor cell luciferase activity was 97.40% with BiCAR-T+OVDual+OVmIL15/21 treatment vs 89.47% with BiCAR-T+OVDual+OVmIL15/21 treatment in PBT022 cells at E:T of 0.5 and 24 h post-treatment ($p < 0.0001$) (Fig. 4H). In contrast, BiCAR-T cells exhibited no statistically significant difference against GBM cells infected with OVmIL15/21 or OVGFP in the absence of OVDual (Fig. S9A). For example, % reduction in tumor cell luciferase activity with BiCAR-T+OVmIL15/21 vs BiCAR-T+OVGFP was 33.88% vs 33.93% in PBT022 cells at E:T of 0.5 and 24 h post-treatment ($p > 0.05$).

- Though you include BiCAR-T alone, a condition with uninfected GBM + BiCAR-T + OVmIL15/21 (no antigen delivery) would confirm that the cytokine effect alone doesn't drive non-specific killing.

Response: We have now included uninfected GBM (no antigen delivery) + BiCAR-T + OVmIL15/21 (BiCAR-T+OVmIL15/21) as suggested and compared its effect with uninfected GBM (no antigen delivery) + BiCAR-T + OVGFP (BiCAR-T + OVGFP). We detected no statistically significant difference in tumor killing between the two conditions (Fig. S9A). Similarly, no differences were observed between control T + OVDual + OVmIL15/21 and control T + OVDual + OVGFP (Fig. S9B). In contrast, enhanced killing was detected with the treatment of BiCAR-T+OVDual+OVmIL15/21, compared to BiCAR-T+OVDual+OVGFP (Fig. 4H). These findings indicate that the cytokines alone do not induce non-specific killing. Thank you for the suggestion.

- Report both statistical significance and actual % differences in killing (e.g., 35% vs. 25%) to contextualize the magnitude of enhancement.

Response: We have now provided both statistical significance and actual % differences in killing in the text when describing the results of Fig. 4H. See above.

- Use brackets or legend notes to clarify that asterisks indicate comparisons between OVmIL15/21 and OVGFP groups.

Response: We have now used brackets for asterisks to show the comparison between OVmIL15/21 and OVGFP groups in revised Fig. 4H.

Figures 5 & S3: These figures show OV spread, T cell presence, and microglial activation via immunofluorescence. However, all data are qualitative.

- Quantify % area or mean intensity of vaccinia, hCD45, IBA1, and CD68 signals per brain section.
- Include statistical comparisons across treatment groups.

Response: We have now quantified % area of vaccinia+ and hCD45+ signals in revised Fig. S7 (previous Fig. S3) and % area of vaccinia+, hCD45+, IBA1+, and CD68+IBA1+ signals per brain section in revised Fig. 5 and included statistical comparison across treatment groups (Fig. 5 and Fig. S7).

Figure 7: The workflow is clear, but BiCAR signaling in NK cells is not functionally validated.

- After antigen stimulation, measure CD107a, Granzyme B, or IFN- γ production to confirm BiCAR-NK functional activation.

Response: We have now shown activation of downstream signaling, including CD69, CD107a, and IFN- γ in BiCAR-NK cocultured with GBM cells infected with OVDual (with antigen stimulation), compared to BiCAR-NK cocultured with GBM cells infected with OVGFP or GBM cells without OV infection (without antigen stimulation) (Fig. S12 C and D).

Figure 8: While the combination shows robust tumor control, several mechanistic elements are missing.

- Track BiCAR-NK persistence and tumor infiltration (e.g., via labeled cells or hCD45/CD56 staining in brain sections).

- Quantify tumor burden reduction as fold change or % relative to control.

Response: BiCAR-NK persistence and tumor infiltration is shown in Fig. S13 now. We were able to detect BiCAR-NK 2 weeks and 4 weeks after OV treatment, with increased population of BiCAR-NK cells in the presence of OVmIL15/21 (Fig. S13).

As requested, we have quantified tumor burden reduction relative to the no-treatment control and revised the text as follows: The combination of BiCAR-NK with OVDual and OVmIL15/21 led to dramatically reduced tumor progression, compared to the no treatment control group (e.g. 93.1% vs 0% reduction in tumor luciferase activity relative to no-treatment control at week 2) (Fig. 8B and 8C), indicating the potent anti-tumor efficacy of this combination. Tumor progression in mice treated with BiCAR-NK plus OVDual and OVmIL15/21 was also reduced, compared to that in mice treated with BiCAR-NK plus OVDual and OVGFP (e.g. 93.1% vs 83.2% reduction in tumor luciferase activity relative to no-treatment control at week 2) (Fig. 8B and 8C), indicating that OVmIL15/21 can boost the anti-tumor effect of the combination of BiCAR-NK with OVDual. We also detected reduced tumor progression in mice treated with BiCAR-NK together with OVDual

and OVMIL15/21, compared to that in mice treated with OVDual and OVMIL15/21 without BiCAR (e.g. 93.1% vs 68% reduction in tumor luciferase activity relative to no-treatment control at week 2) (Fig. 8B and 8C).

Reviewer #4 (Remarks to the Author)-Brain tumor and oncolytic virus expert: I co-reviewed this manuscript with one of the reviewers who provided the listed reports. This is part of the Nature Communications initiative to facilitate training in peer review and to provide appropriate recognition for Early Career Researchers who co-review manuscripts.

Response: We thank the reviewer for the great efforts in reviewing our manuscript and providing constructive suggestions.

In summary, we thank all reviewers for the great efforts in reviewing our manuscript and providing constructive suggestions. We have now addressed your comments and suggestions by performing additional experiments and providing critical controls and mechanistic insights as suggested, which have greatly strengthened our manuscript. We hope the reviewers agree that the manuscript is now much stronger and ready for publication in Nature Communications.

REVIEWERS' COMMENTS

Reviewer #1 (Remarks to the Author):

This is overall a much improved version of the manuscript, with my previous comments addressed, and only a small number of points remaining:

Response: We thank the reviewer for the positive assessment of the revised manuscript and for the remaining constructive suggestions.

Major Point relating to multiple figures (Fig. 5, S7, S13): Pseudoreplication in Statistical Analysis.

It appears that the statistical comparisons have been performed using individual brain slices as independent data points (n=9 per group, derived from 3 slices per mouse across an n of 3 mice). This approach constitutes pseudoreplication and artificially inflates your sample size, leading to an underestimation of variance and potentially spurious statistical significance (low p-values). Your true biological unit of replication is the individual mouse, not the brain slice. Slices taken from the same animal are technical replicates; they are not independent observations as they share the same genetic background, environmental factors, and treatment history. Treating them as independent violates the assumptions of most standard statistical tests and can lead to erroneous conclusions.

To address this critical issue, I strongly recommend the following:

1. **Aggregate Data at the Biological Replicate Level:** For each mouse, calculate a single average value (e.g., percent) for each cellular quantification across its three brain slices. This will provide one biological data point per mouse.
2. **Perform Statistics on Mouse-Level Data:** Conduct all primary statistical analyses using these aggregated mouse-level values. This means your sample size for each experimental group will be $n = 3$ (representing the three individual mice), not $n = 9$. It looks like this is likely still significant for many of the figures, but regardless of significance, it is more accurate.
3. **Adjust Presentation of Data:**
 - o In your figures, if you wish to show the variability within mice (i.e., the individual slice data), please do so in a manner that clearly indicates the nesting (e.g., using distinct symbols or colors for each mouse, or connecting lines between slices from the same mouse). However this does not change the statistical tests. Ensure that the statistical tests and reported n values in your methods and results sections accurately reflect the number of biological replicates (mice; ie 3).

Response: We agree that the individual mouse is the appropriate biological unit of replication. All relevant datasets have therefore been re-analyzed at the mouse level. Specifically, for each animal, measurements from the three brain slices were averaged to generate a single value per mouse. All statistical analyses were re-performed using these aggregated values, resulting in $n = 3$ mice per group. Figures 5, S7, and S13 have been remade accordingly, and figure legends, Methods, and Results sections have been updated to report the n of mice and corresponding statistical analysis.

Minor point: The statement “Although the survival benefit of BiCAR-T cells combined with OVDual was modest in the xenograft NSG model, the implications of this finding are likely far more impactful in the clinical setting” is an overstatement as often these therapies are less impactful in the clinical setting; revise wording to not speculate beyond what the data show.

Response: We have revised the indicated sentence to remove speculative statements regarding clinical impact but simply stating that “Although the survival benefit of BiCAR-T cells combined with OVDual was modest in the xenograft NSG model, the results support further evaluation of this combination in clinical studies”.

Reviewer #2 (Remarks to the Author):

The authors have addressed, although not in full, my previous concerns. I appreciate the effort made to strengthen the manuscript by incorporating the requested in vitro and in vivo controls. These additions certainly improve the technical rigor of the study.

That said, the critical issue remains the very modest therapeutic efficacy observed with the BiCAR approach compared to the mono-CAR treatment (Figure S6). While the multimodal strategy is conceptually interesting and may increase targetability in the context of antigen escape in GBM, the survival benefit demonstrated here is limited. It remains unclear whether this level of improvement justifies the added complexity of the proposed multimodal platform.

Moreover, the authors’ comment “the implications of this finding are likely more impactful in the clinical setting, because NSG mice are immunodeficient and do not recapitulate the complex human GBM tumor microenvironment, which includes diverse immune population, cytokine network, and selective pressure that drive dynamic changes in antigen expression.” remains highly speculative.

The translational path for combining dual-antigen oncolytic viruses with BiCAR-T/NK cells appears challenging. Manufacturing complexity, regulatory burden, and clinical implementation may significantly limit its applicability, particularly in light of the modest efficacy observed. Overall, while the strategy is innovative, stronger evidence of therapeutic advantage would be needed to support its relevance and competitiveness in the GBM cell therapy landscape.

Response: We thank the reviewer for acknowledging the added in vitro and in vivo controls, which have strengthened the rigor of the study. We have revised the indicated sentence by removing the speculative statement regarding clinical impact, “the implications of this finding are likely more impactful in the clinical setting”, but simply stating that “the results support further evaluation of this combination in clinical studies”.

Reviewer #3 (Remarks to the Author):

The authors have answered all my questions

Response: We thank the reviewer for the positive response.

Reviewer #4 (Remarks to the Author):

Response: We thank the reviewer for co-reviewing our manuscript.